# The termination of UHRF1-dependent PAF15 ubiquitin signaling is regulated by USP7 and ATAD5

Ryota Miyashita[1], Atsuya Nishiyama[1]*, Weihua Qin[2], Yoshie Chiba[1], Satomi Kori[3], Norie Kato[3], Chieko Konishi[1], Soichiro Kumamoto[1], Hiroko Kozuka-Hata[4], Masaaki Oyama[4], Yoshitaka Kawasoe[5], Toshiki Tsurimoto[5], Tatsuro S Takahashi[5], Heinrich Leonhardt[2], Kyohei Arita[3], Makoto Nakanishi[1]*

[1]Division of Cancer Cell Biology, The Institute of Medical Science, The University of Tokyo, Tokyo, Japan; [2]Faculty of Biology, Ludwig-Maximilians-Universität München, Munich, Germany; [3]Structural Biology Laboratory, Graduate School of Medical Life Science, Yokohama City University, Yokohama, Japan; [4]Medical Proteomics Laboratory, The Institute of Medical Science, The University of Tokyo, Tokyo, Japan; [5]Laboratory of Chromosome Biology, Department of Biology, Faculty of Science, Kyushu University, Fukuoka, Japan

*For correspondence:
uanishiyama@g.ecc.u-tokyo.ac.jp (AN);
mkt-naka@g.ecc.u-tokyo.ac.jp (MN)

**Competing interest:** The authors declare that no competing interests exist.

**Abstract** UHRF1-dependent ubiquitin signaling plays an integral role in the regulation of maintenance DNA methylation. UHRF1 catalyzes transient dual mono-ubiquitylation of PAF15 (PAF15Ub2), which regulates the localization and activation of DNMT1 at DNA methylation sites during DNA replication. Although the initiation of UHRF1-mediated PAF15 ubiquitin signaling has been relatively well characterized, the mechanisms underlying its termination and how they are coordinated with the completion of maintenance DNA methylation have not yet been clarified. This study shows that deubiquitylation by USP7 and unloading by ATAD5 (ELG1 in yeast) are pivotal processes for the removal of PAF15 from chromatin. On replicating chromatin, USP7 specifically interacts with PAF15Ub2 in a complex with DNMT1. USP7 depletion or inhibition of the interaction between USP7 and PAF15 results in abnormal accumulation of PAF15Ub2 on chromatin. Furthermore, we also find that the non-ubiquitylated form of PAF15 (PAF15Ub0) is removed from chromatin in an ATAD5-dependent manner. PAF15Ub2 was retained at high levels on chromatin when the catalytic activity of DNMT1 was inhibited, suggesting that the completion of maintenance DNA methylation is essential for the termination of UHRF1-mediated ubiquitin signaling. This finding provides a molecular understanding of how the maintenance DNA methylation machinery is disassembled at the end of the S phase.

## Editor's evaluation

Following up the previous observation that UHRF1-mediated dual mono-ubiquitylation of PAF15 (PAF15Ub2) promotes PAF15 chromatin loading and DNMT1 recruitment to the DNA replication sites, this study provides convincing evidence showing that termination of PAF15Ub2 signaling is regulated by USP7-mediated deubiquitylation and ATAD5-mediated removal from chromatin. These are important findings for our understanding of how the maintenance DNA methylation machinery is disassembled post replication.

## Introduction

DNA methylation at CpG dinucleotide is an epigenetic modification that regulates various biological processes, including gene silencing, genome stability, cellular development, and differentiation (*Greenberg and Bourc'his, 2019*; *Jones, 2012*; *Moore et al., 2013*). DNA methylation is stably maintained during cell proliferation (*Jones and Liang, 2009*; *Petryk et al., 2020*). DNA methyltransferase 1 (DNMT1) plays a key role in the maintenance of DNA methylation by catalyzing the conversion of hemi-methylated DNA to a fully methylated state (*Edwards et al., 2017*). In addition, recent studies have also suggested the potential *de novo* function of DNMT1 (*Haggerty et al., 2021*; *Jialun et al., 2020*; *Li et al., 2018*). Besides the C-terminal catalytic domain, DNMT1 contains several regulatory regions, including proliferating cell nuclear antigen (PCNA)-interacting protein motif (PIP-box), replication foci targeting sequence (RFTS), a CXXC zinc finger domain, and two bromo-adjacent homology domains (*Lyko, 2018*). DNMT1 specifically localizes at DNA methylation sites dependently on the RFTS domain, which interacts with dual mono-ubiquitylated histone H3 (H3Ub2) or PAF15 (PAF15Ub2; *Ishiyama et al., 2017*; *Nishiyama et al., 2020*; *Qin et al., 2015*). The RFTS domain of DNMT1 also shows preferential H3K9me3 binding over H3K9me0 to enhance the interaction with H3Ub2 (*Ren et al., 2021*). These interactions also cause release of autoinhibition and enzymatic activation of DNMT1, presumably via the conformational change (*Ishiyama et al., 2017*; *Mishima et al., 2020*; *Syeda et al., 2011*; *Takeshita et al., 2011*; *Zhang et al., 2015*).

Dual mono-ubiquitylation of histone H3 and PAF15 is catalyzed by an E3 ubiquitin ligase ubiquitin-like (UBL) containing plant homeodomain (PHD) and RING finger domains 1 (UHRF1), also known as NP95 or ICBP90 (*Karg et al., 2017*; *Nishiyama et al., 2013*). UHRF1 binds specifically to hemi-methylated DNA via its SET and RING-associated (SRA) domain (*Arita et al., 2008*; *Avvakumov et al., 2008*; *Hashimoto et al., 2008*) and plays an essential role for the DNMT1 recruitment to sites of DNA methylation (*Bostick et al., 2007*; *Sharif et al., 2007*). The E3 ubiquitin ligase activity of UHRF1 is enhanced by binding to hemi-methylated DNA (*Harrison et al., 2016*) and mutations in the RING finger domain, which is responsible for ubiquitin ligase activity, impair the localization of DNMT1 to methylation sites and maintenance DNA methylation (*Nishiyama et al., 2013*; *Qin et al., 2015*). The N-terminal UBL domain promotes interaction with the E2 enzyme, Ubch5/UBE2D (*DaRosa et al., 2018*; *Foster et al., 2018*). The PHD and tandem Tudor (TTD) domains are responsible for recognizing and binding to the N-terminal portion of histone H3 and PAF15 (*Arita et al., 2012*; *Nishiyama et al., 2020*; *Rajakumara et al., 2011*; *Rothbart et al., 2012*). UHRF1 dissociates from chromatin upon conversion of hemi-methylated DNA to fully methylated DNA, leading to the inactivation of UHRF1-dependent ubiquitin signaling (*Nishiyama et al., 2020*).

PAF15 is a PCNA-binding protein (*De Biasio et al., 2015*, *Emanuele et al., 2011*, *Karg et al., 2017*; *Nishiyama et al., 2020*; *Yu et al., 2001*) and transiently binds to chromatin during S phase in PCNA- and DNA replication-dependent manner (*Nishiyama et al., 2020*). PAF15Ub2 specifically binds to the RFTS domain of DNMT1 to facilitate DNMT1-mediated maintenance of DNA methylation mainly at early replicating domains. Whereas H3Ub2 can be induced in late S phase or when PAF15Ub2 is perturbed to compensate the recruitment of DNMT1 and ensure the stable inheritance of DNA methylation (*Nishiyama et al., 2020*). In addition, inhibition of UHRF1-dependent PAF15 ubiquitylation significantly impairs PAF15 chromatin binding (*Karg et al., 2017*; *Nishiyama et al., 2020*), suggesting that ubiquitylation of PAF15 plays an important role not only in its interaction with DNMT1 but also in its own chromatin binding. Given that more than 80% of CpG methylation on the genome is maintained by DNA replication-coupled maintenance (*Charlton et al., 2018*, *Ming et al., 2020*) and that PAF15Ub2 is a key regulator of replication-coupled DNMT1 chromatin recruitment, a regulatory mechanism for PAF15 ubiquitylation is critical for faithful propagation of DNA methylation patterns. It is speculated that inefficient termination of PAF15 ubiquitin signaling will result in overloading of DNMT1 and unregulated DNA methylation, which is frequently observed in various types of tumors. However, it is not fully understood how the termination of PAF15 ubiquitin signaling is regulated during the process of maintenance DNA methylation.

Protein ubiquitylation is a reversible post-translational modification (*Komander and Rape, 2012*). Among nearly 100 deubiquitylating (DUB) enzymes, USP7 (Ubiquitin-Specific Protease 7, also known as HAUSP) has been shown to accumulate at DNA methylation sites in a complex with DNMT1 or UHRF1 (*Felle et al., 2011*; *Ma et al., 2012*; *Qin et al., 2011*; *Yamaguchi et al., 2017*; *Zhang et al., 2015*). While it has been reported that USP7 promotes efficient maintenance of DNA methylation

through stabilization of DNMT1 and UHRF1 by preventing their polyubiquitylation and proteasomal degradation (*Cheng et al., 2015b*; *Du et al., 2010*), recent studies have shown that USP7 also modulates the level of ubiquitylated histone H3 and histone H2B on chromatin (*Jialun et al., 2020*; *Yamaguchi et al., 2017*). However, it remains unclear whether USP7 also regulates the PAF15 ubiquitylation.

In this report, we set out to study the molecular mechanism of PAF15 chromatin unloading to understand how the termination of replication-coupled maintenance DNA methylation is regulated. Using the cell-free system derived from *Xenopus* egg extracts that recapitulate the processes of maintenance DNA methylation, we demonstrate that the unloading of PAF15Ub2 is regulated by the two regulatory mechanisms, namely USP7-dependent deubiquitylation and unloading of PAF15 by ATPase family AAA domain-containing protein 5 (ATAD5). We also find that PAF15 unloading is tightly coordinated with the completion of maintenance DNA methylation and requires the release of UHRF1 from chromatin. Finally, co-depletion of USP7 and ATAD5 from egg extracts results in an elevated global DNA methylation. We propose that timely inactivation of PAF15 is critical for the faithful inheritance of DNA methylation patterns.

## Results

### Identification of USP7 as a PAF15 binding protein

UHRF1-dependent ubiquitin signaling plays a critical role in PAF15 chromatin binding. To test whether the DUB activity is required for termination of PAF15 ubiquitylation signaling, we employed ubiquitin vinyl sulfone (UbVS), a pan DUB inhibitor (*Borodovsky et al., 2001*). In *Xenopus* egg extracts, sperm chromatin added to egg cytoplasm assembles a functional nucleus and undergoes chromosomal replication (*Blow and Laskey, 2016*). As it has been reported that treatment of *Xenopus* egg extracts with UbVS inhibits ubiquitin turnover, resulting in depletion of free ubiquitin (*Dimova et al., 2012*), we also added excess free ubiquitin before incubation with sperm chromatin to activate ubiquitylation pathways. As previously reported, PAF15 underwent dual mono-ubiquitylation on chromatin during S phase and then dissociated from chromatin (*Nishiyama et al., 2020*; *Povlsen et al., 2012*). The addition of UbVS alone significantly delayed PAF15 and DNMT1 chromatin loading, confirming the importance of ubiquitin signaling for the initiation of maintenance of DNA methylation (*Figure 1A*, *Figure 1—figure supplement 1*). In contrast, inhibition of DUB by UbVS plus excess free ubiquitin led to enhanced and prolonged chromatin association of DNMT1 and PAF15. The addition of ubiquitin alone did not significantly affect the level of DNMT1 and PAF15 on chromatin (*Figure 1—figure supplement 1*). These results indicate that the DUB activity was required for the termination of maintenance DNA methylation in egg extracts.

To examine the possibility that PAF15 interacts proteins related to DUB, we performed glutathione-S-transferase (GST)-PAF15 pull-down and nanoflow liquid chromatography-tandem mass spectrometry (nanoLC-MS/MS) for identification of proteins. Sepharose beads bound to GST or GST-PAF15 were incubated with *Xenopus* interphase egg extracts. Beads-bound proteins were eluted by cleavage of GST with thrombin protease (*Figure 1B*). Recovered proteins were identified by nanoLC-MS/MS (*Figure 1C*; *Supplementary files 1 and 2*). This analysis confirmed the PAF15 binding with PCNA and revealed the interaction between PAF15 and USP7. The interaction between GST-PAF15 and USP7 was also validated by immunoblotting using USP7 specific antibodies. Mutations of two phenylalanine to alanine within the consensus PIP sequence of PAF15 abolished the interaction with PCNA, but not USP7, suggesting that the PAF15-USP7 interaction is independent of PCNA (*Figure 1*, *Figure 1—figure supplement 1*). This interaction was further demonstrated by reciprocal immunoprecipitation and western blotting experiments for endogenous proteins in egg extracts using anti-PAF15 and USP7 antibodies (*Figure 1E*). Our results indicate that USP7 has an activity to interact with PAF15 although it was still unclear at this stage whether other protein(s) might be involved in this interaction.

### PAF15 associates with USP7 through the TRAF and UBL2 domains

It has been reported that the binding of USP7 to substrate proteins involves two distinct domains: one is the N-terminal TRAF (TNF-receptor-associated factors-like) domain, and the other is the C-terminal UBL domain (*Al-Eidan et al., 2020*). Previous reports have shown that the recognition of the P/A/ ExxS motif via the binding pocket within the TRAF domain of USP7 is important for the interaction with many substrate proteins such as p53, MDM2, and MCM-BP (*Hu et al., 2006*; *Jagannathan et al.,*

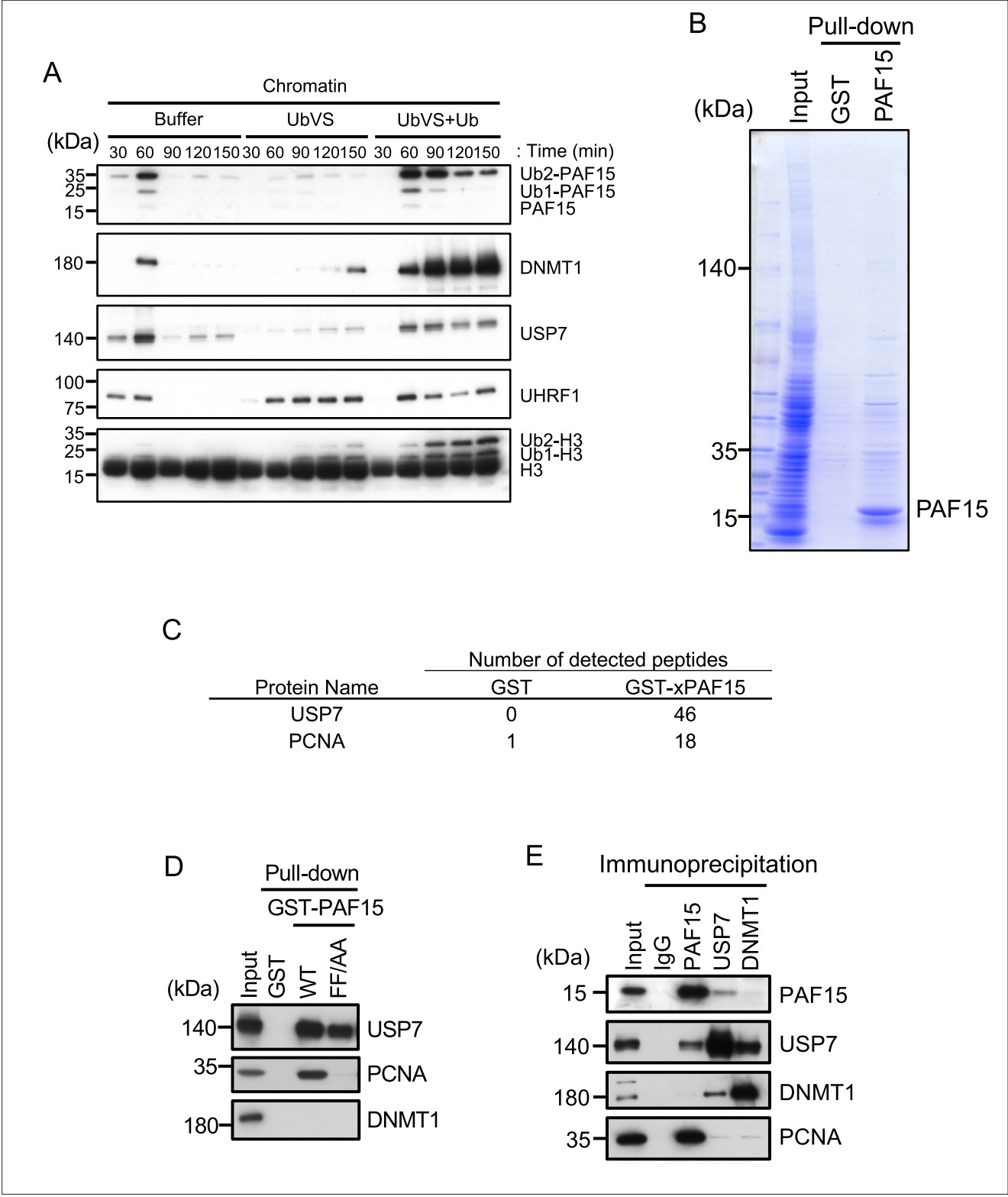

**Figure 1.** USP7 was identified as a PAF15 binding protein. (**A**) Sperm chromatin was added to interphase egg extracts supplemented with either buffer (+Buffer), 20 μM UbVS (+UbVS), or 20 μM UbVS and 58 μM ubiquitin (+UbVS + Ub). Samples were analyzed by immunoblotting using the antibodies indicated. (**B**) Proteins pull-downed from interphase egg extracts by GST and GST-PAF15 were stained by Coomassie Brilliant Blue. (**C**) The samples from GST-PAF15 pull down were analyzed by nanoflow liquid chromatography-tandem mass spectrometry (nanoLC-MS/MS). Selected proteins were indicated in the table. (**D**) GST pull-down assay was performed by GST or GST-PAF15 wild-type (WT) or PIP mutant (FF/AA), and the samples were analyzed by

*Figure 1 continued on next page*

*Figure 1 continued*

immunoblotting using the antibodies indicated. (**E**) Immunoprecipitation was performed by PAF15, USP7, and DNMT1 antibodies-bound beads, and the samples were analyzed by immunoblotting using the antibodies indicated. Source data are provided as *Figure 1—source data 1*.

The online version of this article includes the following source data and figure supplement(s) for figure 1:

**Source data 1.** *Figure 1* Original blots.

**Figure supplement 1.** USP7 was identified as a PAF15 binding protein.

**Figure supplement 1—source data 1.** *Figure 1—figure supplement 1* Original blots.

*2014*; *Sheng et al., 2006*). Meanwhile, it has also been reported that the UBL2 domain recognizes KxxxK motifs that interact with acidic surface patches within the UBL2 (*Cheng et al., 2015a*). We searched for these motifs in PAF15 and found two P/AxxS motifs ([76]PSTS[79] and [94]AGGS[97]) and one KxxxK motif ([101]KKPRK[105], *Figure 2A*). We then tested whether these sequences serve as binding sites for USP7 by mutating serine residues in the P/AxxS motif and two lysine residues in the KxxxK motif to alanine (rPAF15 SA and KA, respectively) and combining these mutations to produce a triple mutant (rPAF15 SAKA). As described above, GST-PAF15 was able to pull down both USP7 and PCNA from *Xenopus* interphase egg extracts (*Figure 2B*). The PIP-box mutant of PAF15 lost interaction with PCNA but retained binding to USP7. In contrast, mutations in the P/AxxS or KxxxK motifs reduced the binding of USP7 to GST-PAF15. Furthermore, the triple mutations completely lost the binding of PAF15 to USP7 but retained the binding with PCNA. These results suggest that USP7 interacts with PAF15 through both the TRAF domain and the UBL2 domain.

Next, we confirmed the requirement of the TRAF and UBL2 domains of USP7 for the interaction with PAF15 by performing pull-down experiments from egg extracts using 3xFLAG-tagged-rUSP7 and its mutants expressed in insect cells (*Figure 2C*). The TRAF domain (residues 1–208) interacted efficiently with PAF15, while the catalytic domain (residues 209–582) and UBL1-5 domain (residues 583–1124) of USP7 did not (*Figure 2D*). Deletion of the TRAF domain significantly impaired the interaction between PAF15 and USP7, although it did not affect the binding to DNMT1. This is further supported by the observation that the introduction of the W167A mutation, which disrupts the TRAF binding pocket (*Sheng et al., 2006*), resulted in a loss of interaction with PAF15. Deletion of the UBL domain or mutations into the UBL2 pocket, D758A/E759A/D764A (*Cheng et al., 2015b*), also decreased binding to PAF15. The PAF15 interaction with the USP7 UBL domain was validated by isothermal titration calorimetry (ITC, *Figure 2—figure supplement 1*). hUSP7$_{561-1102}$ bound to full-length hPAF15 dependently on KxxxK motif with a $K_D$ of 32.7±5.8 μM, which is weaker than the binding of the hUSP7$_{561-1102}$ and DNMT1 ($K_D$: 0.6 μM, *Cheng et al., 2015b*), suggesting that the interaction between PAF15 KxxxK and USP7 UBL2 alone may not be sufficient to compete against the DNMT1-USP7 interaction. Taken together, these results indicate that both the TRAF and UBL2 domains of USP7 contribute to the interaction with PAF15, as has been recently reported for other USP7 substrates (*Ashton et al., 2021*, *Georges et al., 2019*).

## USP7 is involved in PAF15 dissociation from chromatin during S phase progression

We next tested whether USP7 regulates PAF15 on chromatin. To this end, we examined the chromatin binding of a recombinant PAF15 mutant lacking USP7 binding activity in PAF15-depleted extracts. As seen for endogenous PAF15, wild-type rPAF15 dissociated from chromatin at 120 min when added to PAF15-depleted egg extracts (*Figure 3A*, *Figure 3—figure supplement 1*). In contrast, the rPAF15 with mutated USP7 interacting sequences (SAKA) showed prolonged chromatin association even after 120 min, although chromatin unloading of USP7 was delayed. To directly test the importance of USP7 for regulation of PAF15 chromatin binding, we depleted USP7 from egg extracts. Compared to the control, the USP7-depleted extracts showed impaired dissociation of PAF15 chromatin (*Figure 3B*, *Figure 3—figure supplement 1*). Affinity-purified recombinant USP7 efficiently restored PAF15 chromatin dissociation, but the USP7 C225S, C223 in human, catalytic inactive mutant failed to do so (*Hu et al., 2002*; *Li et al., 2002*; *Figure 3B*, *Figure 3—figure supplement 1*). The USP7 specific inhibitor FT671 also suppressed the PAF15Ub2 chromatin dissociation (*Figure 3C*, *Figure 3—figure supplement 1*). These results suggested that USP7 regulates PAF15 chromatin dissociation through its DUB activity. To investigate whether USP7 directly deubiquitylates PAF15, we performed an *in vitro* DUB

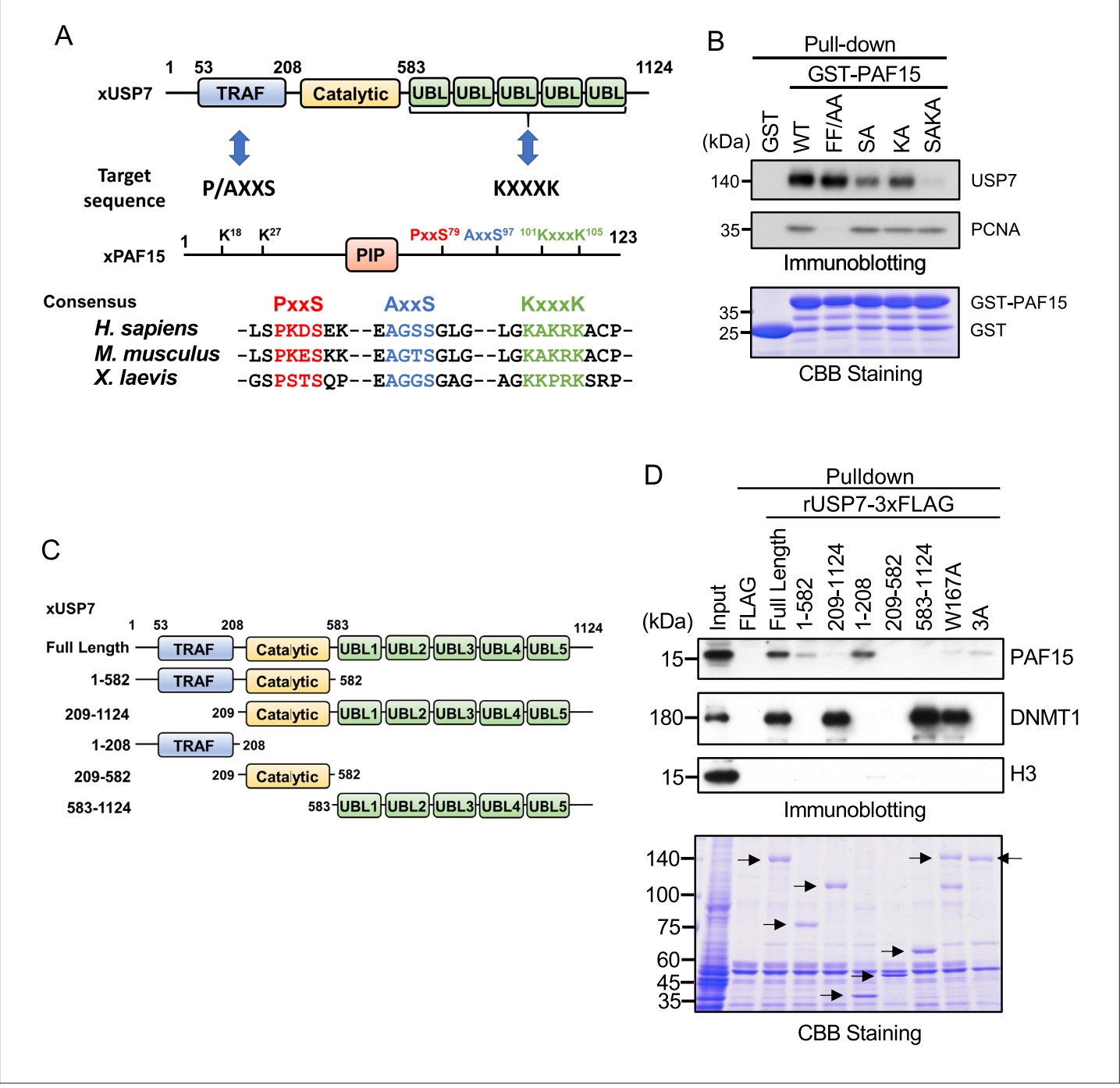

**Figure 2.** PAF15 associates with the TRAF and UBL1-2 domains of USP7. (**A**) Schematic illustration of PAF15-USP7 binding experiment. USP7 recognizes P/A/ExxS or KxxxK motifs in its substrates via TRAF or ubiquitin-like (UBL) domains, respectively. PAF15 has three motifs, and alanine mutations were introduced at S79, S97, K101, and K105. (**B**) GST pull-down from the interphase egg extracts using GST-PAF15, P/AxxS mutant (S79A/S97A; SA), KxxxK mutant (K101A/K105A; KA), and triple mutant (SAKA). The samples were analyzed by immunoblotting using the antibodies indicated. Purified GST or GST-PAF15 mutants used in pull-down assay were stained using CBB. (**C**) Schematic illustration of rUSP7 truncation mutants employed in (**D**). (**D**) FLAG pull-down from interphase egg extracts using rUSP7-3xFLAG mutants presented in (**C**), W167A and 3 A point mutants. USP7 3 A: D780A/E781A/D786A. The samples were analyzed by immunoblotting using the antibodies indicated. Samples were also stained by CBB. Arrowheads indicate rUSP7 truncation mutants and point mutants. Source data are provided as *Figure 2—source data 1*.

The online version of this article includes the following source data and figure supplement(s) for figure 2:

**Source data 1.** *Figure 2* Original blots.

*Figure 2 continued on next page*

*Figure 2 continued*

**Figure supplement 1.** hUSP7$_{561-1102}$ interacts with hPAF15 dependently on KxxxK motif.

**Figure supplement 1—source data 1.** *Figure 2—figure supplement 1* ITC.

assay by using purified recombinant ubiquitylated hPAF15 and hUSP7. We ubiquitylated hPAF15 by incubating with E1 (mouse UBA1), E2 (UBE2D3), and E3 (UHRF1) enzymes *in vitro*. After purification, we incubated ubiquitylated hPAF15 with recombinant hUSP7 and analyzed the reaction products. USP7 WT efficiently deubiquitylated PAF15 while the catalytic inactive USP7 mutant (C223A) did not (*Figure 3D*). USP47 that is functionally related DUB to USP7 showed little DUB activity toward the ubiquitylated PAF15 (*Figure 3—figure supplement 1*). These results suggested that USP7 directly deubiquitylates PAF15 to promote PAF15 chromatin dissociation.

To further elucidate the mechanism underlying termination of PAF15 signaling by USP7, we examined whether USP7 interacts with PAF15 on chromatin. As shown in previous reports (*Nishiyama et al., 2020*), chromatin-bound PAF15 existed mainly as ubiquitylated forms (PAF15Ub2 or PAF15Ub1), and PAF15Ub2 specifically interacted with DNMT1 (*Figure 4A*). USP7 co-immunoprecipitated with PAF15 as well as DNMT1. Importantly, PAF15Ub2 was readily detected in the USP7 immunoprecipitates, whereas PAF15Ub1 and PAF15Ub0 were not. Given that DNMT1 forms a complex with USP7 and predominantly binds to PAF15Ub2, USP7 binding to PAF15 might be mediated by DNMT1 at DNA methylation sites.

In order to inhibit the interaction between DNMT1 and USP7, we introduced mutations into the KG linker of DNMT1, which is responsible for USP7 binding (*Cheng et al., 2015b*; *Yamaguchi et al., 2017*). We performed immunodepletion of endogenous DNMT1 from egg extracts and added-back wild-type DNMT1 or the KG linker mutant (DNMT1 4KA). As previously reported, immunodepletion of DNMT1 inhibited the chromatin binding of USP7 and induced marked accumulation of ubiquitylated PAF15 and histone H3 (*Nishiyama et al., 2020*; *Nishiyama et al., 2013*). Wild-type DNMT1 efficiently restored USP7 chromatin recruitment and PAF15 chromatin dissociation but DNMT1 4KA failed to do so (*Figure 4B and C*, *Figure 4—figure supplement 1*). These results suggest that DNMT1 recruits USP7 to chromatin and mediates the formation of USP7-PAF15Ub2 complex. Consistent with this idea, immunodepletion of PAF15 had no significant effect on the level of chromatin-bound USP7 (*Figure 4D*, *Figure 4—figure supplement 1*).

## Unloading of PAF15 couples with the completion of DNA methylation maintenance

We next tested how PAF15 chromatin dissociation is coordinated with the progression of maintenance DNA methylation. Completion of DNA maintenance methylation is accompanied by conversion of hemi-methylated DNA to fully methylated DNA by DNMT1 and subsequent inactivation of UHRF1-dependent ubiquitin signaling. To inhibit DNMT1 activity, we replaced the endogenous DNMT1 with a recombinant DNMT1 C1101S mutant that lacks DNMT1 catalytic activity (*Sharif et al., 2007*; *Takeshita et al., 2011*; *Wyszynski et al., 1993*). The results showed that the inactivation of DNMT1 led to accumulation of UHRF1 on chromatin, presumably due to the failure in the conversion of hemi-methylated DNA to fully methylated state (*Figure 5A*, *Figure 5—figure supplement 1*). PAF15Ub2 showed a significant accumulation on chromatin along with H3Ub2 under this condition, suggesting that the completion of maintenance DNA methylation is required for the USP7-dependent dissociation of PAF15 from chromatin. Note that the recruitment of USP7 to chromatin was rather enhanced when DNMT1 was inactivated (*Figure 5A*). Immunoprecipitation of PAF15 or USP7 from chromatin lysates showed that inhibition of the catalytic activity of DNMT1 did not affect the binding of USP7 to PAF15Ub2 (*Figure 5B*). These results suggest that the USP7-mediated deubiquitylation couples the completion of DNA methylation by DNMT1.

Failure of DNA methylation replication has been shown to be accompanied by accumulation of UHRF1 on chromatin and enhanced UHRF1-dependent ubiquitin signaling. We hypothesized that when maintenance DNA methylation is inhibited, the enhanced E3 ligase activity of UHRF1 caused by its chromatin accumulation may overcome the DUB activity of USP7, which apparently suppresses the deubiquitylation of PAF15. Recent studies have reported that the maternal gene Stella/DPPA3, which protects against oocyte-specific DNA methylation in mice, binds directly to the UHRF1-PHD

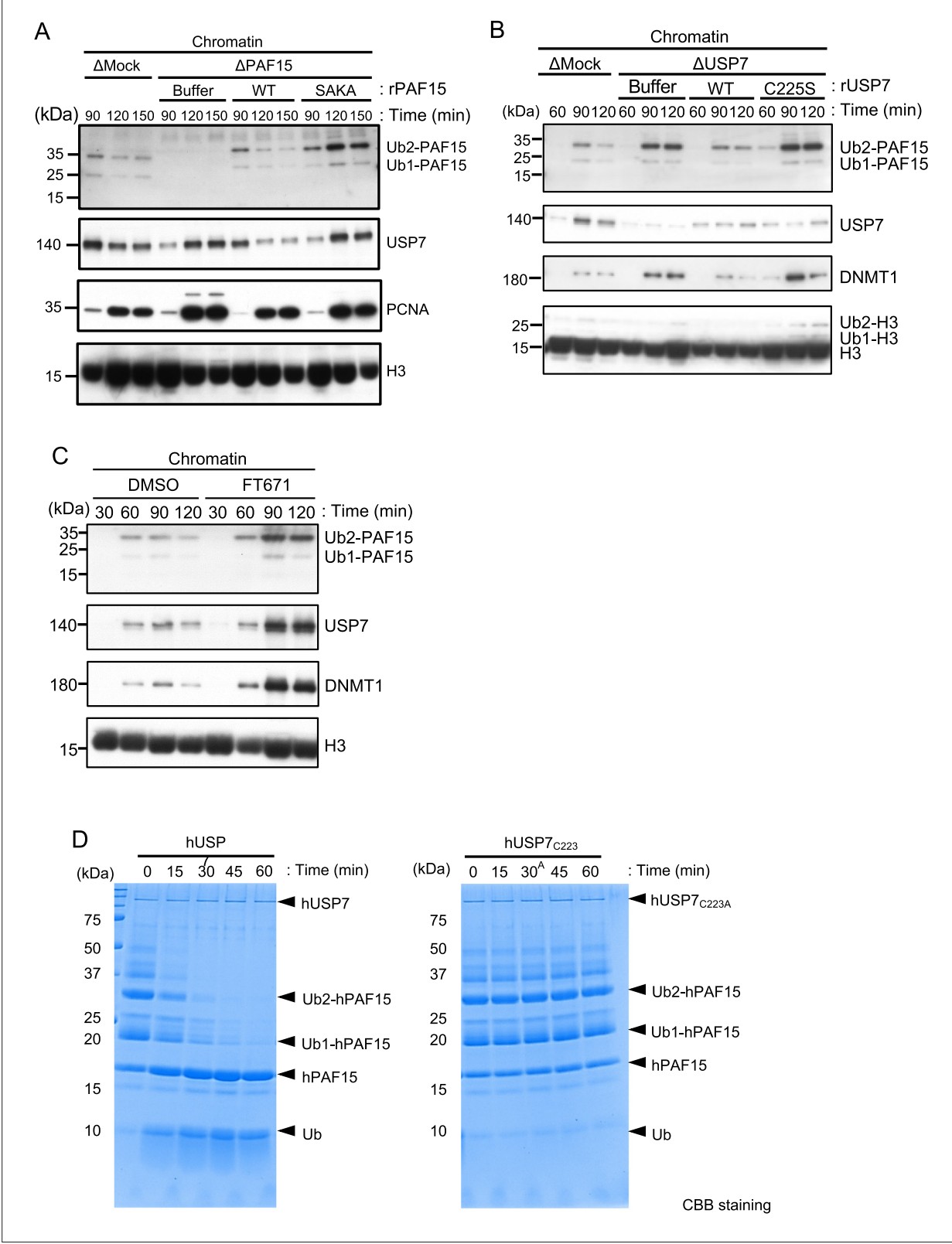

**Figure 3.** USP7 promotes PAF15 dissociation from chromatin. (**A**) Sperm chromatin was added to Mock- or PAF15-depleted interphase extracts supplemented with either buffer (+Buffer), wild-type rPAF15-3xFLAG (+WT) or rPAF15 SAKA-3xFLAG (+SAKA). Chromatin fractions were isolated, and the samples were analyzed by immunoblotting using the antibodies indicated. (**B**) Sperm chromatin was added to Mock- or USP7-depleted interphase extracts supplemented with either buffer (+Buffer), wild-type rUSP7-3xFLAG (+WT) or catalytic mutant rUSP7 C225S-3xFLAG (+C225 S).

*Figure 3 continued on next page*

*Figure 3 continued*

Chromatin fractions were isolated, and the samples were analyzed by immunoblotting using the antibodies indicated. (**C**) Sperm chromatin was added to interphase extracts supplemented with either dimethyl sulfoxide (DMSO) (+DMSO) or FT671 (+FT671). Chromatin fractions were isolated, and the samples were analyzed by immunoblotting using the antibodies indicated. (**D**) Ubiquitylated hPAF15 was incubated with recombinant hUSP7 WT (left) or C223A catalytic mutant (right) at indicated times. The reaction products were analyzed by SDS-PAGE with CBB staining. Source data are provided as *Figure 3—source data 1*.

The online version of this article includes the following source data and figure supplement(s) for figure 3:

**Source data 1.** *Figure 3* Original blots.

**Figure supplement 1.** USP7 directly deubiquitylates PAF15 to promote PAF15 chromatin dissociation.

**Figure supplement 1—source data 1.** *Figure 3—figure supplement 1* Original blots.

domain and inhibits UHRF1 nuclear localization and chromatin binding (*Du et al., 2019*; *Li et al., 2018*; *Mulholland et al., 2020*). We have previously shown that the addition of recombinant mouse DPPA3 to egg extracts inhibits the chromatin-binding activity of UHRF1 and induces dissociation of UHRF1. To determine whether UHRF1 competes with the deubiquitylation by USP7, we forced chromatin dissociation of UHRF1 by adding the purified recombinant GST-mDPPA3 to DNMT1 depleted extracts supplemented with the DNMT1 C1101S mutant (*Figure 5C*). The addition of recombinant mDPPA3 efficiently induced chromatin dissociation of UHRF1, leading to a significant decrease in the levels of chromatin-bound PAF15 and DNMT1 (*Figure 5D*, *Figure 5—figure supplement 1*). Importantly, USP7 depletion caused significant delay of PAF15 chromatin dissociation induced by mDPPA3 addition. These results suggest that UHRF1 maintains PAF15 chromatin association by counteracting USP7-dependent PAF15 deubiquitylation until the completion of maintenance DNA methylation.

## ATAD5 promotes PAF15 unloading from chromatin

It has been shown that PCNA is unloaded from chromatin by the ATAD5-RLC (RFC-like complex) in a coordinated manner with the maturation of Okazaki fragment in the late S phase (*Johnson et al., 2016*; *Kang et al., 2019*; *Kubota et al., 2015*, *Kubota et al., 2013*; *Lee et al., 2010*; *Ulrich and Takahashi, 2013*). Based on the requirement of PCNA for PAF15 chromatin loading described previously, we investigated the role of ATAD5 in the chromatin dissociation of PAF15. Consistent with previous reports in mammalian cultured cells and budding yeast, immunodepletion of ATAD5 from interphase egg extracts resulted in the chromatin accumulation of PCNA (*Figure 6A*; *Kubota et al., 2013*; *Lee et al., 2013*). Interestingly, chromatin binding of PAF15Ub0 was readily detected on ATAD5-depleted chromatin (*Figure 6A*, *Figure 6—figure supplement 1*). On the other hand, no significant change was observed in the amount of PAF15Ub2 on chromatin in ATAD5-depleted extracts. In USP7/ATAD5 co-depleted extracts, PAF15 showed accumulation on chromatin regardless of its ubiquitylation state. The accumulation of the PAF15Ub0 and Ub1 were rescued by recombinant hATAD5-RFCs addition to ATAD5-depleted extracts (*Figure 6B*, *Figure 6—figure supplement 1*). These results suggested that ATAD5 regulates PAF15Ub0 and Ub1 chromatin dissociation.

We next examined whether ATAD5 regulates chromatin unloading of non-ubiquitylated PAF15. First, we added a PAF15 mutant lacking the ubiquitylation sites (PAF15 KRKR) to the PAF15/ATAD5 double-depleted extract and examined its chromatin binding. As expected, PAF15 KRKR did not show chromatin binding in the presence of ATAD5, but its chromatin binding became detectable in ATAD5-depleted extracts (*Figure 6C*, *Figure 6—figure supplement 1*). We also inhibited PAF15 ubiquitylation by UHRF1 depletion. As previously reported, UHRF1 depletion completely inhibited PAF15 ubiquitylation and chromatin loading, resulting in inhibition of DNMT1 recruitment (*Nishiyama et al., 2020*). However, obvious chromatin binding of non-ubiquitylated PAF15 was observed in UHRF1/ATAD5 double-depleted extracts (*Figure 6D*, *Figure 6—figure supplement 1*). These results suggest that non-ubiquitylated or deubiquitylated PAF15 is unloaded in an ATAD5-dependent manner.

## USP7 and ATAD5 promote dissociation of chromatin-bound PAF15

Next, we investigated whether USP7 and ATAD5 accelerates PAF15 chromatin dissociation. To induce PAF15 chromatin accumulation, sperm chromatin was incubated in USP7/ATAD5 co-depleted extracts. Chromatin was isolated after 90 min replication and further incubated with recombinant USP7 WT or C225S catalytic inactive mutant. PAF15Ub2, but not PAF15Ub1 or PAF15Ub0, was

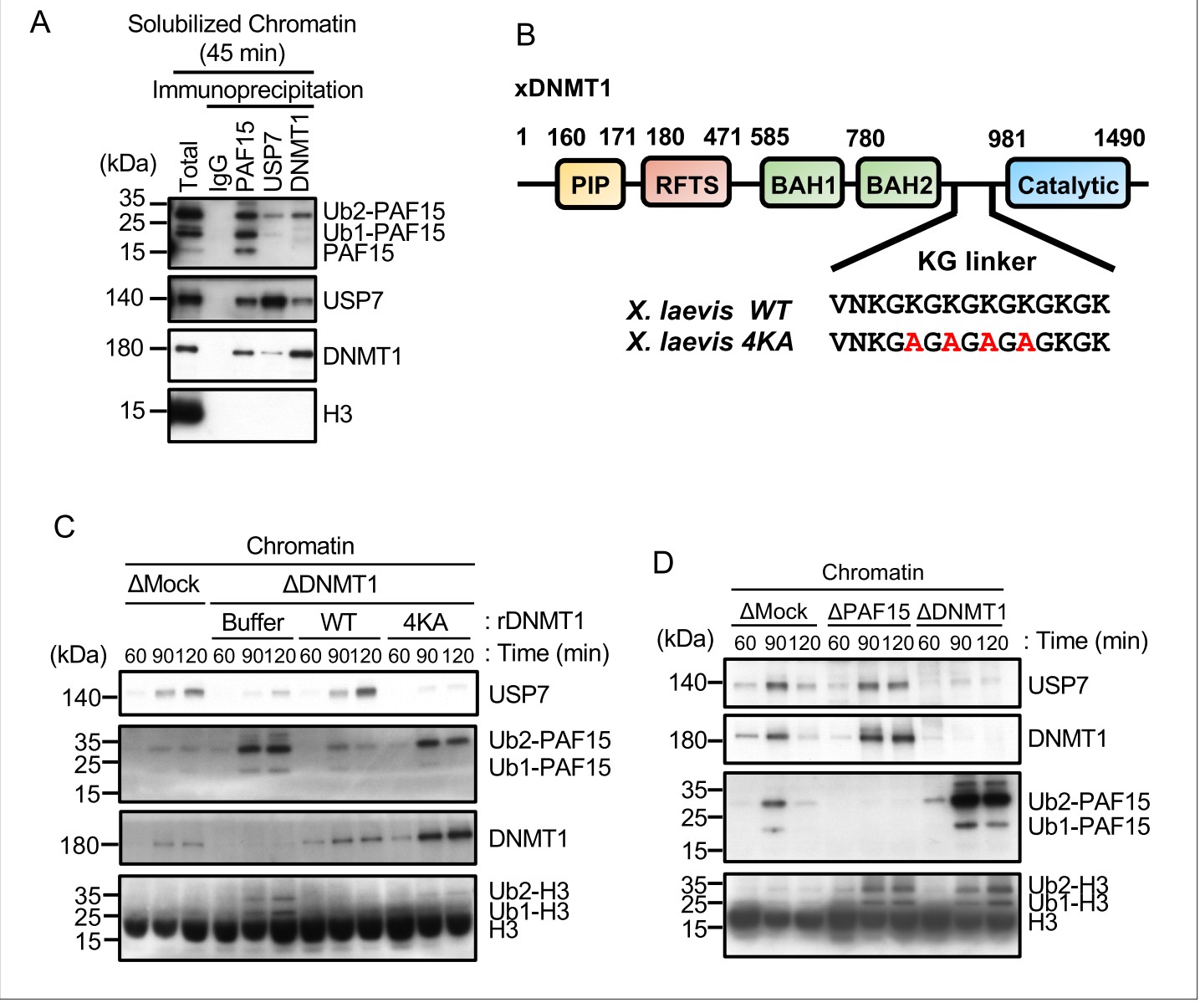

**Figure 4.** USP7 is recruited to chromatin through the interaction with DNMT1 for PAF15 deubiquitylation. (**A**) Sperm chromatin was added to interphase extracts. Replicating chromatin was digested by micrococcal nuclease (MNase). Immunoprecipitation was performed by PAF15, USP7, and DNMT1 antibodies from the solubilized chromatin fraction, and the samples were analyzed by immunoblotting using the antibodies indicated. (**B**) Illustration of DNMT1 domain structure. The KG linker located between the bromo-adjacent homology (BAH) domain and catalytic domain contributes to interaction with USP7. rDNMT1 4KA mutant, using in (**C**), was introduced mutation at four lysines to alanine in the KG linker. (**C**) Sperm chromatin was added to Mock- or DNMT1-depleted interphase extracts supplemented with either buffer (+Buffer), wild-type rDNMT1-3xFLAG (+WT), or rDNMT1 4KA-3xFLAG (+4 KA). Chromatin fractions were isolated, and the samples were analyzed by immunoblotting using the antibodies indicated. (**D**) Sperm chromatin was added to Mock-, PAF15-, and DNMT1-depleted extracts. Chromatin fractions were isolated, and the samples were analyzed by immunoblotting using the antibodies indicated. Source data are provided as *Figure 4—source data 1*.

The online version of this article includes the following source data and figure supplement(s) for figure 4:

**Source data 1.** *Figure 4* Original blots.

**Figure supplement 1.** USP7 is recruited to chromatin through the interaction with DNMT1.

**Figure supplement 1—source data 1.** *Figure 4—figure supplement 1* Original blots.

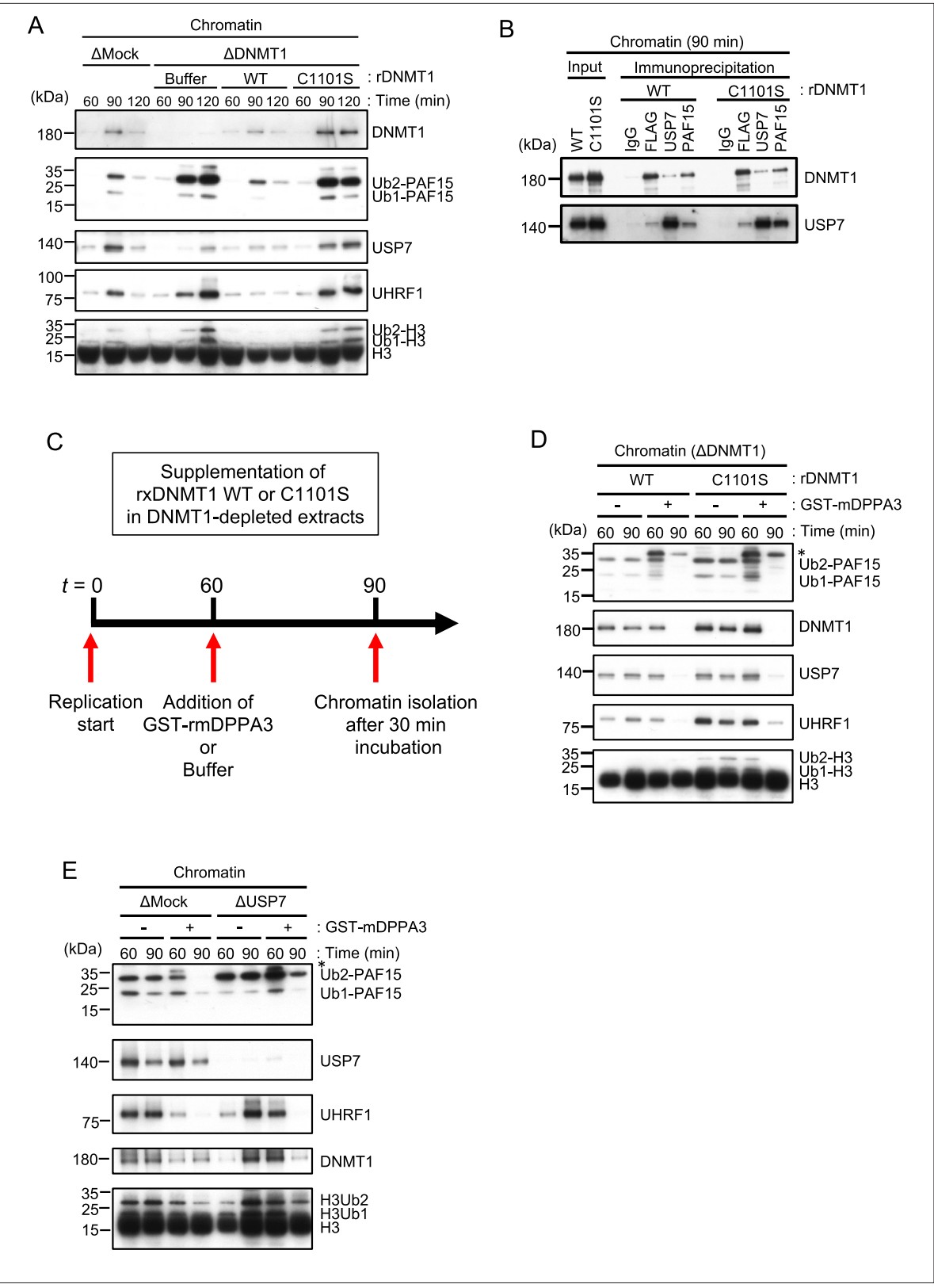

**Figure 5.** Unloading of PAF15 requires DNMT1-dependent DNA methylation. (**A**) Sperm chromatin was added to Mock- or DNMT1-depleted interphase extracts supplemented with either buffer (+Buffer), wild-type rDNMT1-3xFLAG (+WT), or catalytic mutant rDNMT1 C1101S-3xFLAG (+C1101 S). Chromatin fractions were isolated, and the samples were analyzed by immunoblotting using the antibodies indicated. (**B**) Sperm chromatin was added to DNMT1-depleted interphase extracts supplemented wild-type rDNMT1-3xFLAG (+WT) or catalytic mutant rDNMT1 C1101S-3xFLAG (+C1101 S).

*Figure 5 continued on next page*

*Figure 5 continued*

Replicating chromatin was digested by micrococcal nuclease (MNase). Immunoprecipitation was performed by PAF15, USP7 antibodies-bound beads, and FLAG affinity beads in the solubilized chromatin fraction solution, and the samples were analyzed by immunoblotting using the antibodies indicated. (**C**) A schema of an experiment described in D. (**D**) Sperm chromatin was added to DNMT1-depleted interphase extracts supplemented wild-type rDNMT1-3xFLAG (+WT) or catalytic mutant rDNMT1 C1101S-3xFLAG (+C1101 S). After 60 min, the extracts were supplemented with either buffer (−) or GST-mDPPA3 61–150 (+). Chromatin fractions were isolated, and the samples were analyzed by immunoblotting using the antibodies indicated. The asterisk indicates a non-specific band. (**E**) Sperm chromatin was added to USP7-depleted interphase extracts. After 90 min, the extracts were supplemented with either buffer (−) or GST-mDPPA3 61–150 (+). Chromatin fractions were isolated, and the samples were analyzed by immunoblotting using the antibodies indicated. The asterisk indicates a non-specific band. Source data are provided as *Figure 5—source data 1*.

The online version of this article includes the following source data and figure supplement(s) for figure 5:

Source data 1. *Figure 5* Orignal blots.

Figure supplement 1. Unloading of PAF15 requires DNMT1-dependent DNA methylation.

Figure supplement 1—source data 1. *Figure 5—figure supplement 1* Original blots.

efficiently dissociated from chromatin only when recombinant wild-type USP7 was added (*Figure 6E*, *Figure 6—figure supplement 1*). Conversely, when recombinant hATAD5-RLCs was added to USP7/ATAD5 co-depleted extracts after 120 min replication, PAF15Ub1 and Ub0 dissociated from chromatin (*Figure 6F*, *Figure 6—figure supplement 1*). These results suggested that chromatin dissociation of PAF15Ub2 is regulated by USP7, whereas PAF15Ub1 and PAF15Ub0 are regulated by ATAD5-dependent unloading.

## Inhibition of PAF15 chromatin unloading leads to an increase in global DNA methylation

Inhibition of chromatin unloading of PAF15 might affect maintenance DNA methylation. To investigate changes in the level of global DNA methylation and efficiency of DNA replication by USP7- and/or ATAD5-depletion, we measured the incorporation of radiolabeled S-adenosyl-methionine ($^3$H-SAM) and ($\alpha$-$^{32}$P) dCTP into DNA, respectively. USP7/ATAD5 double-depletion caused increased DNA methylation compared to mock-depleted extracts without significant effect on gross DNA replication (*Figure 7*, *Figure 7—figure supplement 1*). DNMT1 immunoprecipitation from chromatin lysates showed enhanced DNMT1 interaction with PAF15Ub2, but not H3Ub2, in USP7/ATAD5 double-depleted extracts compared to mock-depleted extracts (*Figure 7—figure supplement 1*). Either USP7- or ATAD5-depletion alone did not disrupt DNA methylation maintenance in *Xenopus* egg extracts. These data suggest that the termination of PAF15 ubiquitin signaling suppresses excessive DNA methylation.

## USP7 promotes dissociation of chromatin-bound PAF15 to assure a complete DNA methylation in mouse embryonic stem cells

To investigate the interaction between murine USP7 (mUSP7) and murine PAF15 (mPAF15) and the regulation of chromatin association, we used CRISPR/Cas-based gene editing to introduce S75A, S88A, K92A, and K96A mutations into the endogenous *Paf15* gene (mPAF15 SAKA) in wild-type J1 (wt) mouse embryonic stem cells (mESCs; *Figure 8—figure supplement 1*). These cells were first transiently transfected with a GFP-mUSP7 expression construct and used for co-immunoprecipitation (Co-IP) experiments with mPAF15 antibodies. We then performed Co-IP experiments to check the interaction between endogenous mUSP7 and mPAF15 in wt and mPAF15 SAKA mESCs (*Figure 8*). Quantifying the corresponding bands of GFP-mUSP7 and mUSP7 in western blots from both Co-IP experiments shows a significantly reduced binding of mPAF15 SAKA with mUSP7 compared to mPAF15 wt (*Figure 8A*).

To investigate the effect of these mutations on chromatin association, we detected mPAF5 wt and SAKA in different cell fractions by PAF15 IP (*Figure 8B*). Notably, we observed more Ub2-mPAF15 SAKA in the chromatin fraction than Ub2-mPAF15 wt (*Figure 8B*) that is in consistence with our results from *Xenopus* (*Figure 8B*). To assess the influence of the SAKA mutation on the DNA methylation maintenance, we analyzed the DNA methylation levels of LINE-1 elements at the naïve mESCs and epiblast-like cells (EpiLCs). We indeed observed increases of DNA methylation in both mPAF15 SAKA

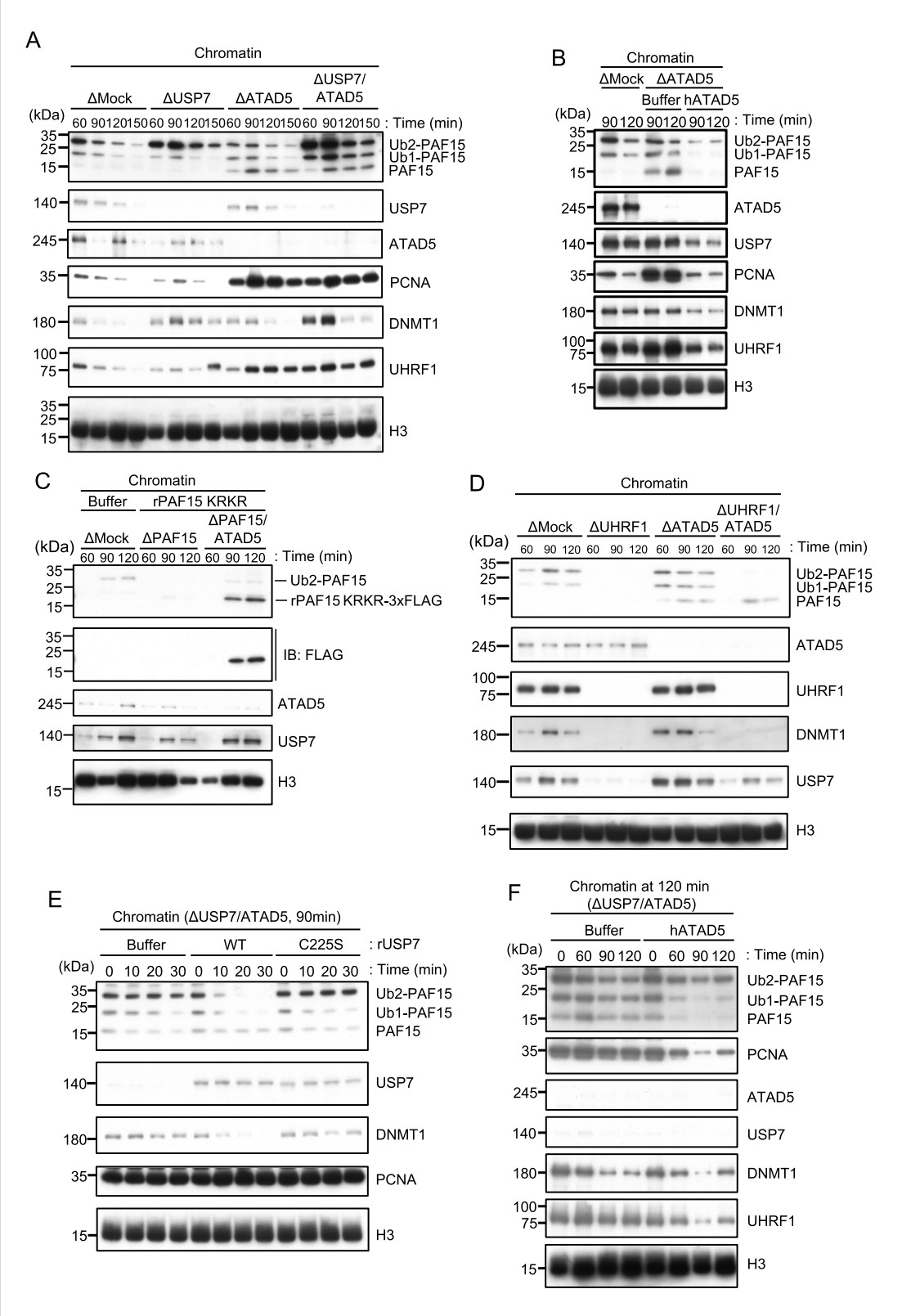

**Figure 6.** ATAD5 unloads PAF15Ub0 from chromatin. (**A**) Sperm chromatin was added to Mock-, USP7-, ATAD5-, and USP7/ATAD5-depleted interphase extracts and isolated at indicated time points. Chromatin bound proteins were analyzed by immunoblotting. (**B**) Sperm chromatin was added to Mock-, ATAD5-depleted interphase extracts supplemented with either buffer (+Buffer) or recombinant hATAD5-RFCs (+ATAD5). Chromatin fractions were isolated, and the samples were analyzed by immunoblotting using the antibodies indicated. (**C**) Recombinant PAF15 K18R/K27R-3xFLAG was

*Figure 6 continued on next page*

*Figure 6 continued*

supplemented to PAF15- and PAF15/ATAD5-depleted extracts, and chromatin fractions were isolated. Chromatin bound proteins were confirmed by immunoblotting. (**D**) Sperm chromatin was added to Mock-, UHRF1-, ATAD5-, and UHRF1/ATAD5-depleted extracts and isolated at indicated time point. Chromatin bound proteins were analyzed by immunoblotting. (**E**) Sperm chromatin was added to USP7/ATAD5-depleted extracts and isolated at 90 min. The chromatin was supplemented with either buffer (+Buffer), USP7 WT-3xFLAG (+WT), or USP7 C225S-3xFLAG (+C225 S) and re-isolated at indicated time points. Chromatin bound proteins were analyzed by immunoblotting. (**F**) Sperm chromatin was added to Mock- and USP7/ATAD5-depleted extracts. After replication at 90 min, the extracts were supplemented with either buffer (+Buffer) or recombinant hATAD5-RFCs (+ATAD5). Chromatin fractions were isolated, and the samples were analyzed by immunoblotting using the antibodies indicated. Source data are provided as *Figure 6—source data 1*.

The online version of this article includes the following source data and figure supplement(s) for figure 6:

**Source data 1.** *Figure 6* Original blots.

**Figure supplement 1.** ATAD5 unloads PAF15Ub0 from chromatin.

**Figure supplement 1—source data 1.** *Figure 6—figure supplement 1* Original blots.

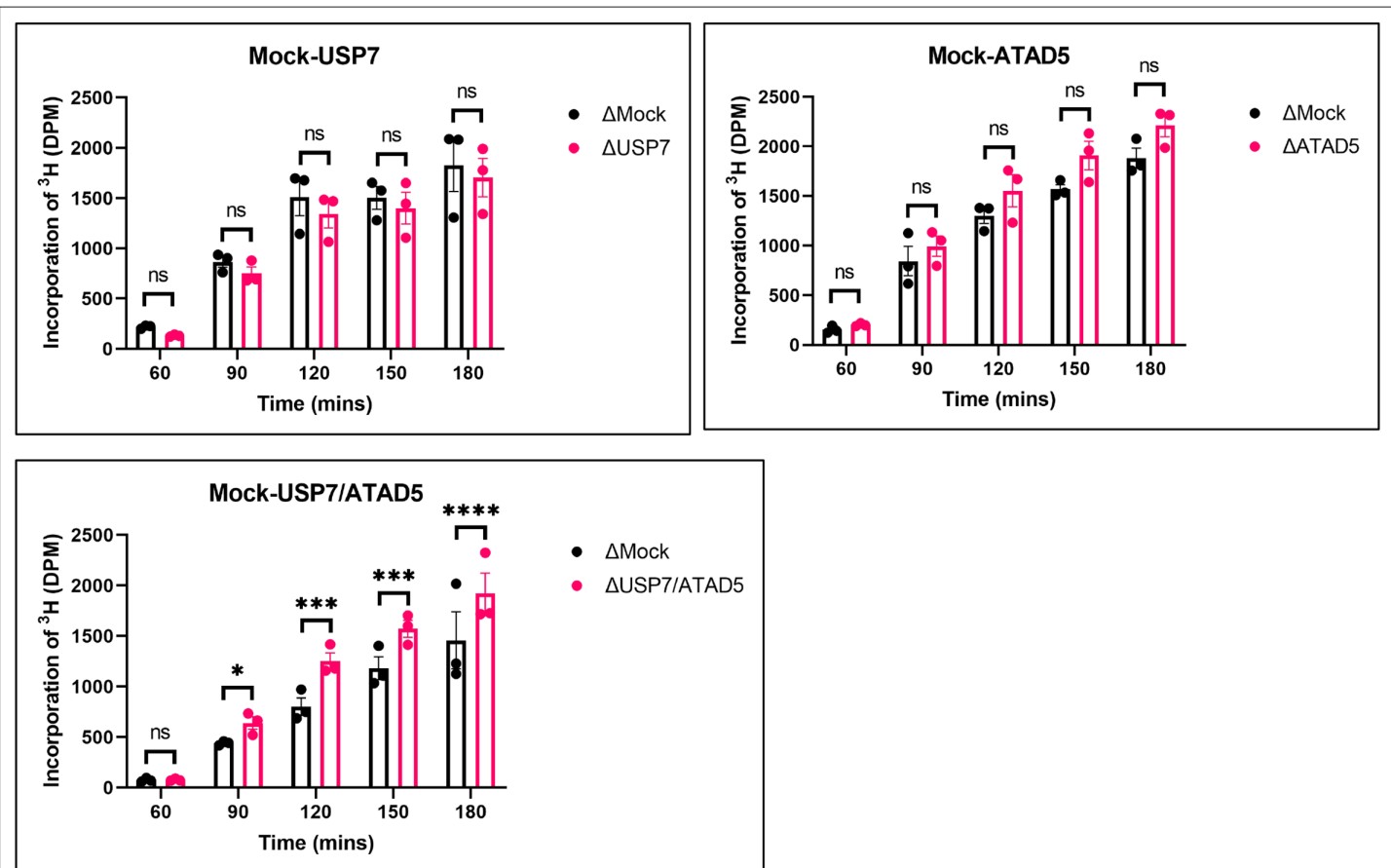

**Figure 7.** PAF15 dissociation negatively regulates aberrant increase of DNA methylation. Sperm chromatin and radiolabeled S-(methyl-$^3$H)-adenosyl-L-methionine were added to either Mock- and USP7-, ATAD5-, or USP7/ATAD5 co-depleted extracts. Purified DNA samples were analyzed to determine the efficiency of DNA methylation. Data are presented as mean ± SEM from three biological replicates. Multiple comparisons were performed by two-way repeated measure ANOVA (RM ANOVA) followed by Sidak's multiple comparison test. ns; not significant, *p<0.05, ***p<0.001, and ****p<0.0001. Source data are provided as *Figure 7—source data 1*.

The online version of this article includes the following source data and figure supplement(s) for figure 7:

**Source data 1.** *Figure 7* DNA methylation.

**Figure supplement 1.** DNA replication and methylation analysis in USP7 or ATAD5-depleted extracts.

**Figure supplement 1—source data 1.** *Figure 7—figure supplement 1* Original blots, DNA replication, and methylation data.

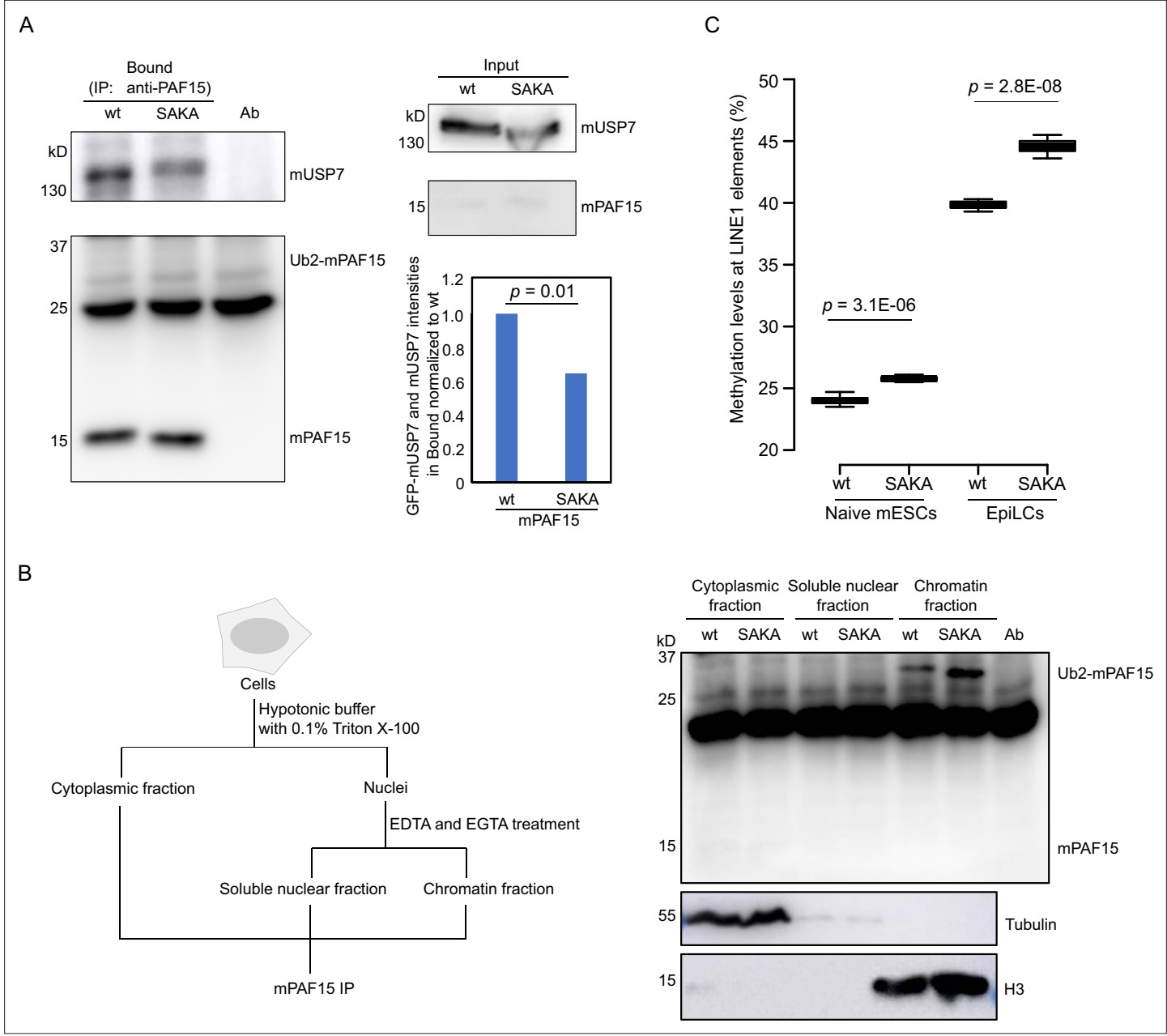

**Figure 8.** mUSP7 interacts with mPAF15 to assure a complete DNA methylation maintenance. (**A**) Immunoprecipitation (IP) of endogenous mPAF15 from whole cell lysates of wt and mPAF15 SAKA mouse embryonic stem cells (mESCs) using an anti-PAF15 antibody. Bound fractions were subjected to immunoblotting with anti-USP7 and PAF15 antibodies. The bar plot shows the quantifications of the relative GFP-mUSP7 and mUSP7 co-precipitated with mPAF15. The error bar stands for SD from three biological replicates. A paired t-test with two tails was done, and p value was indicated. (**B**) Scheme of the cell fractionation experiment described in *Figure 7B* (left). IP of endogenous mPAF15 from cytosolic, soluble nuclear, and chromatin fractions using an anti-PAF15 antibody. Bound fractions were subjected to immunoblotting with PAF15 antibody. Immunoblotting with anti-Tubulin and anti-H3 antibodies is used to indicate the cytosolic and chromatin fractions, respectively. (**C**) Boxplot shows the DNA methylation levels of LINE-1 elements in both wt and mPAF15 SAKA naïve and epiblast-like cells (EpiLCs). Center lines show the medians; box limits indicate the 25th and 75th percentiles as determined by R software; whiskers extend 1.5 times the interquartile range from the 25th and 75th percentiles; outliers are represented by dots. Data sets from four biological replicates were tested for significance with an unpaired t-test with one tail was performed, and p values are indicated.

The online version of this article includes the following source data and figure supplement(s) for figure 8:

**Source data 1.** *Figure 8* Original blots and DNA methylation data.

**Figure supplement 1.** Generation and characterization of mouse embryonic stem cell (mESC) lines carrying the SAKA mutants by genome editing.

**Figure supplement 1—source data 1.** *Figure 8—figure supplement 1* Original gel and sequencing data.

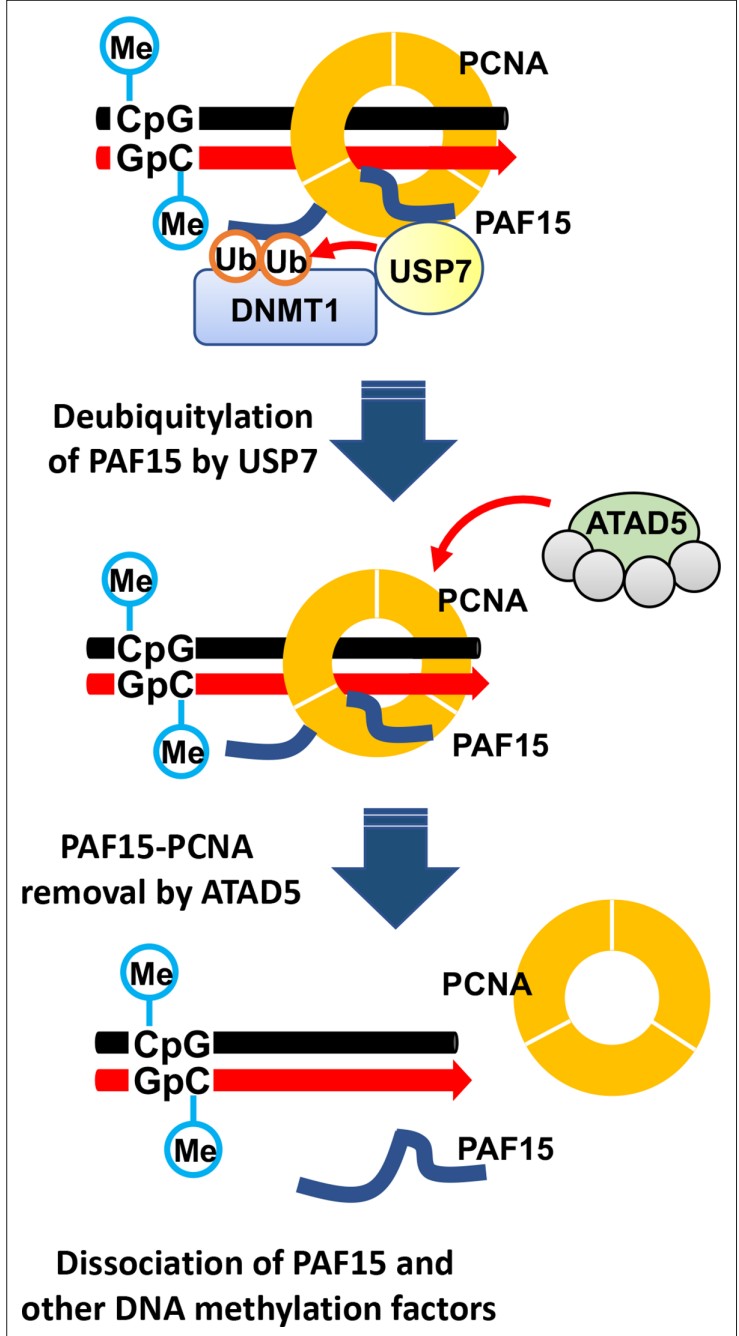

**Figure 9.** Model of the termination of PAF15 ubiquitin signaling post replication.

naïve mESCs and EpiLCs (*Figure 8C*). Taken together, our results suggest that mUSP7 interacts with mPAF15 to assure a complete DNA methylation maintenance in mESCs.

## Discussion

In this study, we investigated how PAF15 chromatin unloading is regulated during the completion of maintenance DNA methylation. Using *Xenopus* egg extracts, we demonstrate that PAF15 unloading is regulated by two distinct mechanisms (*Figure 9*). First, USP7 deubiquitylates PAF15Ub2 to promote PAF15Ub2 dissociation from chromatin. Second, the ATAD5-RLCs complex promotes chromatin unloading of non-ubiquitylated PAF15 and PAF15Ub1 together with PCNA. Importantly, our data

show that co-depletion of USP7 and ATAD5 leads to chromatin accumulation of DNMT1 together with PAF15Ub2 and increased global DNA methylation. Consistent with this data, previous report also showed that the loss of USP7 in HeLa cells leads to the increase of DNA methylation in a substantial fraction of *de novo* DNA methylation sites upon long-term culture (*Jialun et al., 2020*). We speculate that accumulation of both PAF15Ub2 and PCNA on USP7/ATAD5-depleted chromatin causes premature DNMT1 localization and hyperactivation at *de novo* DNA methylation sites.

Our data showed that USP7 specifically targets PAF15Ub2 to facilitate its chromatin dissociation; PAF15Ub2 has been reported to interact with the RFTS domain of DNMT1 (*Nishiyama et al., 2020*), thereby recruiting the DNMT1-USP7 complex to the methylation sites. Since the interaction between USP7 and PAF15 in the cell is probably very weak, the formation of a stable USP7-DNMT1 and PAF15Ub2 complex might be critical for the specific recognition of PAF15Ub2 by USP7. Notably, neither removal nor inhibition of USP7 enhanced histone H3 ubiquitylation, suggesting that PAF15 ubiquitin signaling is the primary pathway to maintain DNA methylation during S phase as previously reported (*Nishiyama et al., 2020*). Although our results indicate that USP7 functions as the major DUB for PAF15 deubiquitylation, it remains possible that other DUBs also influence this process. Indeed, the gradual decrease in chromatin binding of PAF15Ub2 in the absence of USP7 suggests that other proteins are also involved.

Our results also demonstrate that the ATAD5-RLC complex is required for chromatin dissociation of non-ubiquitylated PAF15 and PAF15Ub1. Notably, many of DNA replication proteins interacting with PCNA compete for binding surfaces between ATAD5 and PCNA during DNA replication (*Kang et al., 2019*). We speculate that PAF15 may not interfere with the interaction between ATAD5 and PCNA due to its small size and flexible structure. Our results suggested that PAF15Ub2 is not targeted by ATAD5-dependent unloading. This is consistent with our previous report that dual mono-ubiquitylation of PAF15 plays a pivotal role in its chromatin binding (*Nishiyama et al., 2020*). However, it is not clear how dual mono-ubiquitylation of PAF15 contributes to stable PAF15 chromatin binding. Interestingly, several studies reported the DNA binding activity of ubiquitin. K63-linked ubiquitin chain binds to DNA directly through its DNA interacting patch, consists with threonine, lysine, and glutamic acid (*Liu et al., 2018*). Another paper also has shown that mono-ubiquitylation of transcription factors, such as p53 or IRF1 enhanced their nuclear localization and chromatin binding (*Landré et al., 2017*). Future biochemical analyses will be required to test whether dual mono-ubiquitylation enhances DNA binding activity of PAF15. Alternatively, DNMT1, which forms a complex with PAF15Ub2, may prevent ATAD5-dependent unloading by interacting with PCNA via the PIP-box. Intriguingly, the chromatin binding of non-ubiquitylated PAF15 in ATAD5-depleted extracts did not require UHRF1. These data suggest that PCNA-mediated loading of PAF15 could occur at sites outside DNA methylation sites. In such regions, ATAD5 may prevent the formation of the maintenance DNA methylation machinery by excluding the PAF15-PCNA complex from chromatin.

In summary, data presented here suggest that the coupled ubiquitylation and deubiquitylation may be necessary for proper maintenance of DNA methylation. Interestingly, inactivation of DNMT1 catalytic activity almost completely suppressed chromatin dissociation of PAF15. How is the PAF15 inactivation coupled to the completion of methylation for DNA maintenance? Even when DNA methylation was inhibited, chromatin recruitment of USP7 and the formation of the USP7-PAF15Ub2 complex were observed. Thus, the inhibition of deubiquitylation in this condition is not caused by suppressing USP7 chromatin recruitment or USP7 interaction with PAF15Ub2. Our results suggest that ubiquitylation by UHRF1 is predominant over the deubiquitylation and unloading of PAF15, maintaining PAF15Ub2 until the completion of maintenance methylation by DNMT1. UHRF1 is thought to dissociate from DNA upon binding to hemi-methylated DNA by DNMT1 (*Arita et al., 2008*). Dissociation of UHRF1 from chromatin upon the conversion of hemi-methylated DNA to fully methylated DNA would trigger the removal of ubiquitin moieties from PAF15 to USP7. It is also possible that the binding of DNMT1 to hemi-methylated DNA induces conformational changes in USP7 or PAF15Ub2 to facilitate the deubiquitylation of PAF15Ub2 by USP7. Detailed analysis of the DNMT1-USP7-PAF15Ub2 complex will be important in future studies.

# Methods

**Key resources table**

| Reagent type (species) or resource | Designation | Source or reference | Identifiers | Additional information |
|---|---|---|---|---|
| Cell line (insect cells) | Sf9 | This paper | | Cell line maintained in M. Nakanishi Lab |
| Cell line (mouse cells) | mESC J1 line | This paper | | Cell line maintained in H. LeonhardtLab |
| Antibody | Anti-*Xenopus* PAF15 (Rabbit polyclonal) | PMID:32145273 | PMID:32145273 | WB (1:500) Nakanishi Lab |
| Antibody | Anti-*Xenopus* DNMT1 (Rabbit polyclonal) | PMID:24013172 | PMID:24013172 | WB (1:500) Nakanishi Lab |
| Antibody | Anti-USP7 (Rabbit polyclonal) | Thermo Fisher Scientific | Cat# A300-033A, RRID:AB_203276 | WB (1:1000) |
| Antibody | Anti-*Xenopus* UHRF1 (Rabbit polyclonal) | PMID:24013172 | | WB (1:500) Nakanishi Lab |
| Antibody | Anti-histone H3 (Rabbit polyclonal) | abcam | Cat# ab1791, RRID:AB_302613 | WB (1:3000) |
| Antibody | Anti-PCNA (mouse monoclonall) | Santa Cruz Biotechnology | Cat# sc-56, RRID:AB_628110 | WB (1:1000) |
| Antibody | Anti-*Xenopus* ATAD5 (rabbit polyclonal) | This study | | WB (1:1000) |
| Antibody | Anti-tubulin (mouse monoclonal) | Sigma-Aldrich | Cat# T9026, RRID:AB_477593 | WB (1:2000) |
| Antibody | Anti-PAF15 (mouse monoclonal) | Santa Cruz Biotechnology | Cat# sc-390515 | WB (1:500) |
| Antibody | HRP-anti-mouse IgG (rabbit polyclonal) | Sigma-Aldrich | Cat# A9044, RRID:AB_258431 | WB (1:5000) |
| Recombinant DNA reagent | pGEX4T-3-xPAF15 | PMID:32145273 | PMID:32145273 | Nakanishi Lab |
| Recombinant DNA reagent | pVL1392-xUSP7-3xFLAG | This study | | Expression and purification of xUSP7 in insect cells |
| Recombinant DNA reagent | pVL1392-xDNMT1-3xFLAG | PMID:29053958 | | Nakanishi Lab |
| Recombinant DNA reagent | pGEX-4T-3-mDPPA3 | PMID:33235224 | | Nakanishi Lab |
| Recombinant DNA reagent | pGEX-6P-1-hUSP7 (561–1102) | This study | | Expression and purification of hUSP7 fragment in bacteria cells |
| Recombinant DNA reagent | pGEX-4T-1-SUMO-hPAF15-FLAG | This study | | Expression and purification of full-length hPAF15 in bacteria cells |
| Recombinant DNA reagent | pCSII-EF-mini-AzamiGreen ATAD5 | This study | | Expression and purification of full-length hATAD5 in human 293T cells |
| Recombinant DNA reagent | pCSII-EF-RFC2, 3, 4 and 5 | This study | | Expression and purification of full-length hRFC complex in human 293T cells |
| Peptide, recombinant protein | Ubiquitin | R&D systems (Boston biochem) | U-100H | 58 µM |
| Peptide, recombinant protein | Recombinant Human Ubiquitin Vinyl Sulfone Protein | R&D systems (Boston biochem) | U-202 | 20 µM |
| Commercial assay or kit | EZ DNA Methylation-Gold Kit | Zymo | D5005 | |
| Chemical compound and drug | FT-671 | MedChem Express | HY-107985 | 100 µM |

## Primers

All oligonucleotide sequences are listed in the *Supplementary file 3*.

### *Xenopus* egg extracts

*Xenopus laevis* was purchased from Kato-S Kagaku and handled according to the animal care regulations at the University of Tokyo. The preparation of interphase egg extracts, chromatin isolations, UbVS reactions, DNA replication assays, DNA methylation assays, and immunodepletions was performed as described previously (*Kumamoto et al., 2021*; *Nishiyama et al., 2020*). Unfertilized *X. laevis* eggs were dejellied in 2.5% thioglycolic acid-NaOH (pH 8.2) and washed in 1× Marc's Modified Ringer Solution (MMR) (100 mM NaCl, 2 mM KCl, 1 mM $MgCl_2$, 2 mM $CaCl_2$, 0.1 mM EDTA, and 5 mM HEPES-NaOH [pH 7.5]). After activation in 1× MMR supplemented with 0.3 µg/ml calcium ionophore, eggs were washed with Extraction buffer (EB) (50 mM KCl, 2.5 mM $MgCl_2$, 10 mM HEPES-KOH [pH 7.5], and 50 mM sucrose). Eggs were packed into tubes by centrifugation (BECKMAN, Avanti J-E, JS-13.1 swinging rotor) for 1 min at 190× g and crushed by centrifugation for 20 min at 18,973× g. Egg extracts were supplemented with 50 µg/ml cycloheximide, 20 µg/ml cytochalasin B, 1 mM dithiothreitol (DTT), 2 µg/ml aprotinin, and 5 µg/ml leupeptin and clarified by ultracentrifugation (Hitachi, CP100NX, P55ST2 swinging rotor) for 20 min at 48,400× g. The cytoplasmic extracts were aliquoted, frozen in liquid nitrogen, and stored at –80°C. All extracts were supplemented with an energy regeneration system (2 mM ATP, 20 mM phosphocreatine, and 5 µg/ml creatine phosphokinase). 3000–4000 nuclei/µl of sperm nuclei were added and incubated at 22°C. Aliquots (15–20 µl) were diluted with 150 µl chromatin purification buffer (CPB; 50 mM KCl, 5 mM $MgCl_2$, and 20 mM HEPES-KOH [pH 7.6]) containing 0.1% Nonidet P-40 (NP-40), 2% sucrose, 2 mM N-ethylmaleimide (NEM), and 0.1 mM PR-619. After incubation on ice for 5 min, diluted extracts were layered over 1.5 ml of CPB containing 30% sucrose and centrifuged at 15,000× g for 10 min at 4°C. Chromatin pellets were resuspended in 1× Laemmli sample buffer, boiled for 5 min at 100°C, and analyzed by immunoblotting.

### Antibodies and immunoprecipitations/immunodepletions

*Xenopus* ATAD5 (xATAD5) antibodies were raised in rabbits by immunization with a His10-tagged recombinant xATAD5 fragment encoding 1–289 amino acids and used for immunodepletion and immunoblotting. Rabbit polyclonal antibodies raised against PAF15, DNMT1, and UHRF1 have been previously described. Rabbit polyclonal USP7 antibody (A300-033A) was purchased from Bethyl Laboratories. Mouse monoclonal antibody against PCNA (PC-10) was purchased from Santa Cruz Biotechnology. Rabbit polyclonal histone H3 antibody (ab1791) was purchased from Abcam. For immunoprecipitation, 10 µl of Protein A agarose (GE Healthcare) was coupled with 2 µg of purified antibodies or 5 µl of antiserum. The agarose beads were washed twice with CPB buffer containing 2% sucrose. The antibody beads were incubated with egg extracts for 2 hr at 4°C. The beads were washed three times with CPB buffer containing 2% sucrose and 0.1% Triton X-100 and resuspended in 10 µl of 2× Laemmli sample buffer and 20 µl of 1× Laemmli sample buffer. For immunodepletion, 250 µl of antiserum were coupled to 60 µl of recombinant Protein A Sepharose (rPAS, GE Healthcare). Antibodies bound beads were washed three times in CPB and supplemented with 6 µl fresh rPAS. Beads were split into three portions, and 100 µl of extracts were depleted in three rounds at 4°C, each for 1 hr.

### GST pull-down assay in *Xenopus* egg extracts

Recombinant GST or GST-PAF15 proteins were expressed and purified from *Escherichia coli* (BL21-CodonPlus) and immobilized on Glutathione Sepharose 4B resin (GE Healthcare) for 2 hr at 4°C. The beads were incubated with interphase egg extracts for 2 hr at 4°C. The beads were washed four times with CPB containing 2% sucrose and 0.1% Triton X-100. The washed beads were resuspended in 20 µl of 2× Laemmli sample buffer and 20 µl of 1× Laemmli sample buffer, boiled for 5 min at 100°C, and analyzed by immunoblotting.

### Immunoprecipitation of FLAG-USP7

Recombinant 3xFLAG-tagged USP7 proteins were expressed in Sf9 insect cells. These insect cells were collected and suspended in lysis buffer (20 mM Tris-HCl [pH 8.0], 100 mM KCl, 5 mM $MgCl_2$, 10% glycerol, 1% NP-40, 1 mM DTT, 5 µg/ml leupeptin, 2 µg/ml aprotinin, 20 µg/ml trypsin inhibitor, and 100 µg/ml phenylmethylsulfonyl fluoride [PMSF]), followed by incubation on ice for 10 min. Soluble fractions were isolated after centrifugation of the lysate at 15,000× g for 15 min at 4°C. 2 ml of the soluble lysate was incubated with 30 µl of anti-FLAG M2 affinity resins (Sigma-Aldrich) for 2 hr at 4°C.

The protein-bound beads were washed five times with wash buffer (20 mM Tris-HCl [pH 8.0], 100 mM KCl, 5 mM $MgCl_2$, 10% glycerol, 0.1% NP-40, 1 mM DTT, 5 µg/ml leupeptin, 2 µg/ml aprotinin, 20 µg/ml trypsin inhibitor, and 100 µg/ml PMSF) and stored in PBS at 4°C. 10 µl of protein-bound FLAG beads were coupled with 100 µl of *Xenopus* egg extracts diluted fivefold by CPB containing 2% sucrose and incubated for 2 hr at 4°C. The beads were washed three times by CPB containing 2% sucrose and 0.1% Triton X-100, followed by resuspension by 10 µl of 2× Laemmli sample buffer and 20 µl of 1× Laemmli sample buffer.

## Mass spectrometry

The eluted proteins were trypsin-digested, desalted using ZipTip C18 (Millipore), and centrifuged in a vacuum concentrator. Shotgun proteomic analyses of the digested peptides were performed by LTQ-Orbitrap Velos mass spectrometer (Thermo Fisher Scientific) coupled with Dina-2 A nano-flow liquid chromatography system (KYA Technologies). The samples were injected into a 75-µm reversed-phase C18 column at a flow rate of 10 µl/min and eluted with a linear gradient of solvent A (2% acetonitrile and 0.1% formic acid in $H_2O$) to solvent B (40% acetonitrile and 0.1% formic acid in $H_2O$) at 300 nl/min. Peptides were sequentially sprayed from a nanoelectrospray ion source (KYA Technologies) and analyzed by collision-induced dissociation (CID). The analyses were operated in data-dependent mode, switching automatically between MS and MS/MS acquisition. For CID analyses, full-scan MS spectra (from m/z 380–2,000) were acquired in the orbitrap with a resolution of 100,000 at m/z 400 after ion count accumulation to the target value of 1,000,000. The 20 most intense ions at a threshold above 2000 were fragmented in the linear ion trap with a normalized collision energy of 35% for an activation time of 10 ms. The orbitrap analyzer was operated with the 'lock mass' option to perform shotgun detection with high accuracy. Protein identification was conducted by searching MS and MS/MS data against NCBI (National Center for Biotechnology Information) *X. laevis* protein database using Mascot (Matrix Science). Methionine oxidation, protein N-terminal acetylation, and pyro-glutamination for N-terminal glutamine were set as variable modifications. A maximum of two missed cleavages was allowed in our database search, while the mass tolerance was set to three parts per million for peptide masses and 0.8 Da for MS/MS peaks. In the process of peptide identification, we applied a filter to satisfy a false discovery rate lower than 1%.

## *In vitro* DUB assay

Ubiquitylated PAF15, a substrate for the DUB assay, was prepared by *in vitro* ubiquitylation using recombinant mouse UBA1 (E1), human UBE2D3 (E2), human UHRF1 (E3), ubiquitin, and PAF15. N-terminal six histidine tagged E1 was expressed in Sf9 cells using the baculo virus system according to the manufacture's instruction. The protein was purified by TALON affinity (Clontech), HiTrap-Q anion-exchange (Cytiva) and Hiload 26/600 S200 size-exclusion (Cytiva) chromatographies. E2 was expressed in *E. coli* BL21 (DE3) as a GST-fusion protein and purified by GS4B affinity (Cytiva) and Hiload 26/600 S75 size-exclusion chromatographies (Cytiva). UHRF1 was expressed in *E. coli* Rossetta2 (DE3) and purified using GS4B affinity, HiTrap Heparin, and Hiload 26/600 S200 size-exclusion chromatographies. Ubiquitin was expressed in BL21 (DE3) and purified using HiTrap SP anion-exchange (Cytiva) and Hiload 26/600 S75 size-exclusion chromatographies. PAF15 including C-terminal FLAG tag was expressed in *E. coli*, BL21 (DE3) and purified GS4B affinity, HiTrap SP anion-exchange and Hiload 26/600 S75 size-exclusion chromatographies. The ubiquitylation reaction mixture contained 0.4 µM E1, 6 µM E2, 3 µM E3, 600 µM ubiquitin, and 100 µM PAF15 in a ubiquitylation reaction buffer (50 mM Tris-HCl [pH 8.0], 50 mM NaCl, 5 mM $MgCl_2$, 0.1% Triton X-100, and 2 mM DTT). The reaction mixture was incubated at 25°C for overnight.

For *in vitro* DUB assay, recombinant USP7 full-length wild-type/C223A and deletion of TRAF domain were expressed in Rossetta2 (DE3) and purified by GST-affinity, HiTrap Q anion-exchange, and Hiload 26/600 S200 size-exclusion chromatographies. 3.75 pmol (conc.: 50 nM) of USP7 or USP47 (R&D SYSTEMS, E-626–050) and the ubiquitylated PAF15 were incubated in 75 µl reaction solution in a reaction buffer (20 mM Tris-HCl [pH 7.5], 150 mM NaCl, 0.5 mM DTT, and 10% glycerol) at 20°C for 1 hr. The reaction was stopped at indicated times by adding SDS-sample buffer, and the DUB was analyzed by SDA-PAGE.

## Isothermal titration calorimetry

cDNA of hUSP7, residues 561–1102, was sub-cloned into a pGEX-6P-1 plasmid (Cytiva)/GST-hUSP7 was expressed in *E. coli* Rosetta2 (DE3) and purified using GS4B affinity, HiTrap Q HP anion exchange, and HiLoad 26/600 Superdex 75 size-exclusion chromatography (Cytiva). C-terminal FLAG tagged full-length hPAF15 was expressed as GST-SUMO fusion protein using modified pGEX4T-1 plasmid (Cytiva). The protein was expressed in *E. coli* Rosetta2 (DE3) and purified using GS4B affinity, HiTrap SP HP cation exchange, and 26/600 HiLoad 26/600 Superdex 75 size-exclusion chromatography. Microcal PEAQ-ITC (Malvern) was used for the ITC measurements. The purified proteins were dissolved in 10 mM HEPES (pH 7.5) buffer containing 150 mM NaCl and 0.25 mM tris(2-carboxyethyl)phosphine. 20 µM of $USP7_{561-1102}$ solution in the calorimetric cell was titrated with 1.2 mM of PAF15 solution at 293 K. The data were analyzed with Microcal PEAQ-ITC analysis software using a one-site model. For each interaction, at least three independent titration experiments were performed to show the dissociation constants with the mean SD.

## Recombinant proteins expression and purification

GST-PAF15, 3xFLAG-tagged PAF15, 3xFLAG-tagged DNMT1 WT and 4KA mutant, and GST-mDPPA3 61–150 mutant expression and purification were described previously (*Mulholland et al., 2020*; *Nishiyama et al., 2020*; *Yamaguchi et al., 2017*). S79A/S97A, K101A/K105A mutations in pGEX4T-3, and pKS104-PAF15 constructs were introduced using a KOD-Plus Mutagenesis Kit (Toyobo). These mutant x*PAF15* DNAs from pKS104-PAF15 were amplified by PCR and ligated into pVL1392 vector. *USP7 C225S* mutation was also introduced by KOD-Plus Mutagenesis Kit. GST-tagged protein expression in *E. coli* (BL21-CodonPlus) was induced by the addition of 0.1 M Isopropyl-β-D-1-thiogalactopyranoside to media followed by incubation for 12 hr at 20°C. For purification of GST-tagged proteins, cells were collected and resuspended in lysis buffer (20 mM HEPES-KOH [pH 7.6], 0.5 M NaCl, 0.5 mM EDTA, 10% glycerol, and 1 mM DTT) supplemented with 0.5% NP-40 and protease inhibitors and were then disrupted by sonication on ice. For FLAG-tagged protein expression in insect cells, 3xFLAG-tagged *USP7 WT* or mutants were transferred from pKS103 vector into pVL1392 vector. Baculoviruses were produced using a BD BaculoGold Transfection Kit and a BestBac Transfection Kit (BD Biosciences), following the manufacturer's protocol. Proteins were expressed in Sf9 insect cells by infection with viruses expressing 3xFLAG-tagged PAF15 WT or its mutants for 72 hr at 27°C. Sf9 cells from a 750 ml culture were collected and lysed by resuspending them in 30 ml lysis buffer, followed by incubation on ice for 10 min. A soluble fraction was obtained after centrifugation of the lysate at 15,000× g for 15 min at 4°C. The soluble fraction was incubated for 4 hr at 4°C with 250 µl of anti-FLAG M2 affinity resin equilibrated with lysis buffer. The beads were collected and washed with 10 ml wash buffer and then with 5 ml of EB (20 mM HEPES-KOH [pH 7.5], 100 mM KCl, and 5 mM $MgCl_2$) containing 1 mM DTT. Each recombinant protein was eluted twice in 250 µl of EB containing 1 mM DTT and 250 µg/ml 3xFLAG peptide (Sigma-Aldrich). Eluates were pooled and concentrated using a Vivaspin 500 (GE Healthcare).

The human ATAD5-RFC-like complex (ATAD5-RLC) was expressed and purified as follows. Human 293T cells ($5 \times 10^6$ cells) cultured in a 15cm dish were transfected with 8µg of the human mini-AzamiGreen-tagged *ATAD5* gene, and 1µg each of untagged *RFC2*, *RFC3*, *RFC4*, and *RFC5* genes, all inserted into the pCSII-EF vector, and incubated for 72hr in D-MEM (Sigma-Aldrich) supplemented with 10% fetal bovine serum (FBS) at 37°C. Cells from 10 dishes were harvested, resuspended in 8ml of PBSGE (140mM NaCl, 2.7mM KCl, 10mM $Na_2HPO_4$, 1.7mM $NaH_2PO_4$, 20% glycerol, and 20µM EDTA), supplemented with 1mM PMSF and 20µg/ml leupeptin, and lysed by the addition of 0.5% NP-40. The lysates were incubated on ice for 10min, supplemented with final 0.5M NaCl, and clarified by centrifugation at 75,000× g for 30min at 4°C. Cleared lysates were then applied onto 1ml anti-FLAG M2 affinity resin packed in a column equilibrated with PC buffer (50mM $KPO_4$ [pH 7.5], 0.5mM EDTA, 1mM 3-[(3-Cholamidopropyl)dimethylammonio]propanesulfonate, and 10% glycerol) supplemented with 0.5M NaCl. The column was washed with PC buffer supplemented with 0.5M NaCl, and the ATAD5-RLC proteins were eluted with 100µg/ml FLAG-peptide (Sigma Aldrich) in the same buffer. Peak fractions were collected and used for the assay.

## Quantification of DNA replication and DNA methylation efficiency in *Xenopus* egg extracts

[α-$^{32}$P] dCTP (3000 Ci/mmol) and sperm nuclei were added to interphase extracts and incubated at 22°C. At each time point, extracts were diluted in reaction stop solution (1% SDS, 40 mM EDTA)

and treated with Proteinase K (NACALAI TESQUE, Inc) at 37°C. The solutions were spotted onto Whatman glass microfiber filters followed by 5% trichloroacetic acid containing 2% pyrophosphate. Filters were washed twice in ethanol and dried. The incorporation of radioactivity was counted in the scintillation cocktail. DNA methylation was monitored by the incorporation of S-(methyl-$^3$H)-adenosyl-L-methionine. Extracts supplemented with S-(methyl-$^3$H)-adenosyl-L-methionine and sperm nuclei were incubated at 22°C. At each time point, the reaction was stopped by dilution in CPB containing 2% sucrose up to 300 µl. Genomic DNA was purified using a Wizard Genomic DNA Purification Kit (Promega). Incorporation of radioactivity was counted in the scintillation cocktail.

## Immunoprecipitation from chromatin lysate

MNase-digested chromatin fractions were prepared as described previously (*Nishiyama et al., 2020*). The chromatin pellet was resuspended and digested in 100 µl of digestion buffer (10 mM HEPES-KOH [pH 7.5], 50 mM KCl, 2.5 mM MgCl$_2$, 0.1 mM CaCl$_2$, 0.1% Triton X-100, 2 mM NEM, and 100 µM PR-619) containing 4 U/ml MNase at 22°C for 20 min. The reaction was stopped by the addition of 10 mM EDTA, and the solution was centrifuged at 17,700× g for 10 min. For the immunoprecipitation experiment, 2 µg purified IgG, PAF15, USP7, or DNMT1 antibodies were bound to 10 µl of Protein A agarose beads, and these beads were mixed with digested chromatin lysates at 4°C for 2 hr. After reaction, these beads were washed by CPB containing 2% sucrose and 0.1% Triton X-100, resuspended in 10 µl of 2× Laemmli buffer and 20 µl of 1× Laemmli buffer, and heated at 100°C.

## Cell culture

The mESC J1 line were maintained on 0.2% gelatin-coated dishes in Dulbecco's modified Eagle's medium (Sigma) supplemented with 16% FBS (Sigma), 0.1 mM ß-mercaptoethanol (Invitrogen), 2 mM L-glutamine (Sigma), 1× Minimum Essential Medium non-essential amino acids (Sigma), 100 U/ml penicillin, 100 mg/ml streptomycin (Sigma), recombinant LIF (ESGRO, Millipore), and 2i (1 mM PD032591 and 3 mM CHIR99021 [Axon Medchem, Netherlands]). Cell lines were regularly tested for mycoplasma contamination.

Naïve J1 mouse ESCs were cultured and differentiated into EpiLCs using an established protocol (*Hayashi and Saitou, 2013*; *Qin et al., 2021*). In brief, for both naïve ESCs and EpiLCs, defined media was used, consisting of N2B27: 50% neurobasal medium (Life Technologies), 50% DMEM/F12 (Life Technologies), 2 mM l-glutamine (Life Technologies), 0.1 mM β-mercaptoethanol (Life Technologies), N2 supplement (Life Technologies), B27 serum-free supplement (Life Technologies), 100 U/ml penicillin, and 100 µg/ml streptomycin (Sigma). To differentiate naïve ESCs into EpiLCs, the plates were treated with Geltrex (Life Technologies) at 37°C for 1 hr. Naïve ESCs were plated on Geltrex-treated flasks in defined medium containing 10 ng/ml Fgf2 (R&D Systems), 20 ng/ml Activin A (R&D Systems), and 0.1× Knockout Serum Replacement (Life Technologies). Medium was changed after 24 hr, and EpiLCs were harvested for genomic DNA isolation after 72 hr of differentiation.

## Generation of the PAF15 SAKA mESCs

Genome editing was referred to the previous publication with slight modifications (*Mulholland et al., 2015*). Briefly, the gRNA for editing *Paf15* was designed by using the CRISPR design tool from the Zhang Lab (MIT, http://www.genome-engineering.org/) and was incorporated to pSpCas9 (BB)–2A-Puro (PX459) vector by BpiI restriction sites (*Ran et al., 2013*). To mutate PAF15 wt to the SAKA mutant, a 200nt of donor ssDNA oligo was synthesized by Integrated DNA Technologies. A AluI cutting site was incorporated into the donor oligo for screening. The gRNA and donor oligo were introduced into mESCs by using Lipofectamine 3000 according to the manufacturer's instructions. 48 hr after transfection, cells were plated at colonies density to p100 in the puromycin selection medium (1 µg/ml) for 48 hr. After 8 d, individual clones were picked for genomic DNA isolation. The successful insertion of *Paf15* SAKA mutations was confirmed by Sanger sequencing. DNA oligos used for genome editing are listed in the table.

## Targeted bisulfite amplicon sequencing

According to the manufacturer's instructions, there were 1 × 10$^6$ Naïve and EpiLCs for genomic isolation with QIAamp DNA Mini Kit (Qiagen). 500 ng of gDNA was used for bisulfite conversion followed by the instructions of EZ DNA Methylation-Gold Kit (Zymo), which was eluted in a 2 × 20 µl

Elution Buffer. Targeted bisulfite amplicon sequencing (TaBAseq) was performed as described previously (*Mulholland et al., 2020*). TaBAseq is based on two sequential PCRs. The first one amplifies locus-specific LINE-1 element (chr14, 44537155–44537214), and the second one indexes the sample-specific amplicon with Ilumina's Truseq and Nextera compatible overhangs. The sequencing data was analyzed with a TABSAT package (*Pabinger et al., 2016*).

## Co-IP from mESCs

For Co-IP of USP7, $1.5 \times 10^7$ of mESCs were lysed in 350 µl of lysis buffer (10 mM Tris/Cl pH7.5, 150 mM NaCl, 0.5 mM EDTA, 0.5% NP40, 1.5 mM MgCl$_2$, 0.5 µg/ml Benzonase [Sigma-Aldrich], 1 mM PMSF, 1× mammalian Protease Inhibitor Cocktail [e.g. Serva], and 5 mM NEM [Sigma]) at 4°C for 30 min. Lysates were cleared by centrifugation at 20,000× g for 15 min at 4°C, and the protein concentration was measured using Pierce 660 nm Protein Assay Reagent according to the manufacturer's instructions. Equal amounts of protein extracts were incubated with 8 µl of PAF15 antibodies (Santa Cruz, sc-390515) for 2 hr at 4°C under constant rotation. Then, 20 µl of Magna ChIP Protein A+G Magnetic Beads (Sigma, 16–663) were added and incubated at 4°C under constant rotation for overnight. The bound fractions on the beads were washed three time with washing buffer (10 mM Tris/Cl pH7.5, 150 mM NaCl, and 0.5 mM EDTA) and boiled in Laemmli buffer at 95°C for 10 min. Bound fractions were separated and visualized as a western blot.

Western blots for USP7 were performed using a polyclonal antibody (Bethyl, A300-034A, 1:2000). The antibody used for PAF15 detection were mouse anti-PAF15 antibody (Santa Cruz, sc-390515, 1:500 dilution). The following secondary antibodies conjugated to horseradish peroxidase were used: goat polyclonal anti-rabbit IgG (Bio-rad) and rabbit polyclonal anti-mouse IgG (Sigma, A9044, 1:5000). For detection of horseradish peroxidase-conjugated antibodies, an ECL Plus reagent (GE Healthcare, Thermo Scientific) was used.

To fractionate cells, $1.5 \times 10^7$ of mESCs were treated with 350 µl of hypotonic buffer (10 mM HEPES [pH 7.9], 10 mM KCl, 1.5 mM MgCl$_2$, 0.34 M sucrose, 10% glycerol, 1 mM DTT, 1× Protease Inhibitor, 2 mM PMSF, 5 mM NEM, and 0.1% Triton X-100) at 4°C for 5 min. The cytoplasmic fraction was separated from nuclei by centrifugation at 1300× g for 10 min at 4°C. Nuclei were washed once with the hypotonic buffer and resuspended in the buffer containing 3 mM EDTA, 0.2 mM EGTA, 1 mM DTT, and protease inhibitors as described above at 4°C for 30 min. The nuclear soluble fraction was collected by centrifugation (4 min, 1,700× g, 4°C). The insoluble chromatin was resuspended and incubated at 37°C for 10 min in RIPA buffer supplemented benzonase and protease inhibitors as above. All fractions were supplemented with 150 mM NaCl and clarified by centrifugation at 20,000× g for 15 min at 4°C for PAF15 IP. The fractionation was checked by a polyclonal rabbit-anti-H3 (Abcam, ab1791, 1:5000 dilution) and a monoclonal mouse-anti-tubulin (Sigma, T9026, 1:2000 dilution).

## Statistical analysis

The normal distribution of the population at the 0.05 level was calculated using the Shapiro-Wilk normality test. Data are presented as mean ± SEM, unless otherwise noted. Multiple comparisons were performed by two-way repeated measure ANOVA followed by Sidak's multiple comparison test. For consistency of comparison, significance in all figures is indicated as follows: $*p<0.05$, $**p<0.01$, $***p<0.001$, and $****p<0.0001$.

## Acknowledgements

We thank Chikahide Masutani for human USP7 antibodies. The study is funded by MEXT/JSPS KAKENHI (JP19H05740 to MN; JP19H03143 and JP19H05285 to AN; JP16H06578 to MO; JP19H05741 to KA; 20H03186 and 20H05392 to TS T; 19K16042 to YK). AN was supported in part by a grant from Daiichi Sankyo Foundation of Life Science.

# Additional information

## Funding

| Funder | Grant reference number | Author |
|---|---|---|
| KAKENHI | JP19H05740 | Makoto Nakanishi |
| KAKENHI | JP19H03143 | Atsuya Nishiyama |
| KAKENHI | JP19H05285 | Atsuya Nishiyama |
| KAKENHI | JP16H06578 | Masaaki Oyama |
| KAKENHI | JP19H05741 | Kyohei Arita |
| KAKENHI | 20H03186 | Tatsuro S Takahashi |
| KAKENHI | 20H05392 | Tatsuro S Takahashi |
| KAKENHI | 19K16042 | Yoshitaka Kawasoe |
| Daiichi Sankyo Foundation of Life Science | | Atsuya Nishiyama |

The funders had no role in study design, data collection and interpretation, or the decision to submit the work for publication.

## Author contributions

Ryota Miyashita, Formal analysis, Investigation; Atsuya Nishiyama, Conceptualization, Data curation, Formal analysis, Supervision, Funding acquisition, Writing - original draft; Weihua Qin, Formal analysis, Validation, Investigation; Yoshie Chiba, Satomi Kori, Norie Kato, Chieko Konishi, Soichiro Kumamoto, Hiroko Kozuka-Hata, Masaaki Oyama, Yoshitaka Kawasoe, Toshiki Tsurimoto, Tatsuro S Takahashi, Kyohei Arita, Investigation; Heinrich Leonhardt, Formal analysis, Supervision, Validation, Investigation; Makoto Nakanishi, Conceptualization, Supervision, Funding acquisition, Project administration, Writing - review and editing

## Author ORCIDs

Ryota Miyashita http://orcid.org/0000-0003-1452-8829
Atsuya Nishiyama http://orcid.org/0000-0002-8416-3776
Tatsuro S Takahashi http://orcid.org/0000-0002-1947-7680
Kyohei Arita http://orcid.org/0000-0002-9762-8405
Makoto Nakanishi http://orcid.org/0000-0002-6707-3584

## Decision letter and Author response

Decision letter https://doi.org/10.7554/eLife.79013.sa1
Author response https://doi.org/10.7554/eLife.79013.sa2

# Additional files

## Supplementary files
• MDAR checklist
• Supplementary file 1. MS-based quantification of GST interacting protein.
• Supplementary file 2. MS-based quantification of xPAF15 interacting proteins.
• Supplementary file 3. Oligonucleotides used in this study.

## Data availability

All data generated or analysed during this study are included in the manuscript and supporting files; source data files for all figures have been provided.

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
