## [Editor Report]

Following up the previous observation that UHRF1-mediated dual mono-ubiquitylation of PAF15 (PAF15Ub2) promotes PAF15 chromatin loading and DNMT1 recruitment to the DNA replication sites, this study provides convincing evidence showing that termination of PAF15Ub2 signaling is regulated by USP7-mediated deubiquitylation and ATAD5-mediated removal from chromatin. These are important findings for our understanding of how the maintenance DNA methylation machinery is disassembled post replication.

---

## [Decision Letter]

**Decision letter after peer review:**

Thank you for submitting your article "The termination of UHRF1-dependent PAF15 ubiquitin signaling is regulated by USP7 and ATAD5" for consideration by *eLife*. Your article has been reviewed by 3 peer reviewers, including Xiaobing Shi as Reviewing Editor and Reviewer #1, and the evaluation has been overseen by Jessica Tyler as the Senior Editor.

Essential revisions:

1. Figure 1A – with only a chromatin fraction shown, the authors cannot claim that 'PAF15 underwent dual mono-ub on chromatin and then dissociated from chromatin.' More generally, the paper relies exclusively on this chromatin association western blotting assay for querying chromatin interaction dynamics. Orthogonal approaches should be used to strengthen conclusions being drawn.

2. One of the key observations in this study is the direct interaction between PAF15 and USP7. On one hand, their mutational analyses suggest that this interaction is mediated by both the TRAF-P/AxxS and UBL12-KxxxK contacts, which guided their choices of mutations in subsequent assays. On the other hand, no appreciable binding between PAF15 KxxxK motif and USP7 UBL12 was observed in the context of isolated protein fragments. I find this an important point that needs be further clarified. Note that the interaction between PAF15 KxxxK and USP7 UBL12, if exists, would imply that the USP7-PAF15 interaction may compete against the DNMT1-USP7 interaction, which challenges the notion that DNMT1 strengthens the USP7-PAF15 interaction at methylation sites. Therefore, the authors may want to perform quantitative measurements of the interactions, such as ITC or FP, to clarify the interaction mechanisms between PAF15 and USP7.

3. The overall model is that PAF15 ubiquitylation promotes DNMT1 chromatin association through DNMT1-PAF15ub2 interaction (Figure 8). However, in Figure 1A, under the UbVS+Ub condition, the chromatin bound DNMT1 increases while PAF15ub2 decreases over time. And in Figure 4D, depletion of PAF15 does not affect the levels of chromatin bound DNMT1. These results seem to argue against their model.

4. For such striking effects of these perturbations on DNMT1 chromatin association, it is surprising that the effects on DNA methylation are subtle (Figure 7). This brings into an important question that to what extent does PAF15Ub2 support DNA methylation maintenance? Does it really matter in a physiologic context and for the faithful propagation of DNA methylation patterns through mitotic cell divisions? To strengthen the impact of this work, it will be important to expand DNA methylation analyses and extend findings from *Xenopus* extracts to mammalian cells.

[Editors’ note: further revisions were suggested prior to acceptance, as described below.]

Thank you for resubmitting your work entitled "The termination of UHRF1-dependent PAF15 ubiquitin signaling is regulated by USP7 and ATAD5" for further consideration by *eLife*. Your revised article has been evaluated by Jessica Tyler (Senior Editor) and a Reviewing Editor.

The manuscript has been improved but there are some remaining issues that need to be addressed, as outlined below. Please provide additional experimental data and modify your conclusions carefully reflecting the statistics of the data.

1. Data in Figure 7A shows disruption of USP7 alone is not sufficient to disrupt the rate of DNA methylation maintenance. Data in Figure 7 cannot claim that the observed increase in DNA methylation in USP7/ATAD5-depleted extracts is related to PAF15. Furthermore, the rate of DNA replication in these doubly deleted extracts also looks faster than mock depletion (Supp Figure 7B).

2. The new DNA methylation analysis in Figure 8C analyzing mESCs with disrupted PAF15-USP7 interaction does not show statistically significant differences in EpiLCs.

3. The analysis of newly performed ITC data lacks sufficient clarity. The ITC curve for PAF15 WT in Figure 2—figure supplement 1 does not appear to reflect a 1:1 stoichiometric binding. What was the N value when the Kd of 32.7 µM was determined? If the N value is substantially deviated from 1, an explanation would be warranted. In addition, in the method section, should "26/600 HiLoad 26/600 Superdex 75" be "HiLoad 26/600 Superdex 75"?

---

## [Author Response]

Essential revisions:1. Figure 1A – with only a chromatin fraction shown, the authors cannot claim that 'PAF15 underwent dual mono-ub on chromatin and then dissociated from chromatin.' More generally, the paper relies exclusively on this chromatin association western blotting assay for querying chromatin interaction dynamics. Orthogonal approaches should be used to strengthen conclusions being drawn.

This point is important and well taken. Western blotting analysis of isolated chromatin is a well-established method studying chromatin-binding dynamics of target proteins during DNA replication in *Xenopus* egg extracts. As *Xenopus* egg extracts contain high concentrations of proteins as maternal stock, and only a small fraction of proteins bind to chromatin, it is difficult to distinguish proteins dissociated from chromatin from unbound stock proteins in the supernatants. To strengthen our conclusions with an independent approach, we have performed experiments using mouse ES cells expressing PAF15 mutants. Please see our response to major concern 4 for details.

2. One of the key observations in this study is the direct interaction between PAF15 and USP7. On one hand, their mutational analyses suggest that this interaction is mediated by both the TRAF-P/AxxS and UBL12-KxxxK contacts, which guided their choices of mutations in subsequent assays. On the other hand, no appreciable binding between PAF15 KxxxK motif and USP7 UBL12 was observed in the context of isolated protein fragments. I find this an important point that needs be further clarified. Note that the interaction between PAF15 KxxxK and USP7 UBL12, if exists, would imply that the USP7-PAF15 interaction may compete against the DNMT1-USP7 interaction, which challenges the notion that DNMT1 strengthens the USP7-PAF15 interaction at methylation sites. Therefore, the authors may want to perform quantitative measurements of the interactions, such as ITC or FP, to clarify the interaction mechanisms between PAF15 and USP7.

This point is important and well taken. We have now added data to Supplementary Figure S2 showing that hUSP7 561-1102 (UBL1-5) binds to hPAF15 dependently on KxxxK motif with a binding affinity of 32.7 ± 5.8 μM. As the DNMT1-USP7 interaction shows higher binding affinity (approximately 1 μM) (Cheng et al., Nat.commun, 2015), the interaction between PAF15 KxxxK and USP7 UBL12 may not be sufficient to compete against the DNMT1-USP7 interaction. This result is also in good agreement with our results that the USP7-TRAF domain is important for binding to PAF15. We describe these results in the revised Figure 2—figure supplement 1 and discussed in the main text (page 12).

3. The overall model is that PAF15 ubiquitylation promotes DNMT1 chromatin association through DNMT1-PAF15ub2 interaction (Figure 8). However, in Figure 1A, under the UbVS+Ub condition, the chromatin bound DNMT1 increases while PAF15ub2 decreases over time. And in Figure 4D, depletion of PAF15 does not affect the levels of chromatin bound DNMT1. These results seem to argue against their model.

We apologize that we have not been clear enough explaining why the loss of PAF15Ub2 apparently does not affect the levels of chromatin bound DNMT1. As we have previously reported (Nishiyama et al., Nat.commun, 2020), PAF15Ub2 and H3Ub2 functions in the different context. For the DNMT1 chromatin recruitment, UHRF1 promotes PAF15 ubiquitylation during early S phase, but prefers histone H3 as its substrate in late S phase. Importantly, H3Ub2 also increases when PAF15 ubiquitylation is inhibited, presumably as a back-up mechanism, as shown in Figure 4D. UbVS/Ub treatment (Figure 1A) also induced histone H3 ubiquitylation, suggesting that DNMT1 is localized to chromatin via H3Ub2 under this condition. In the revised manuscript, we have now included a description of the compensation of maintenance DNA methylation via histone H3 ubiquitylation (page 6).

4. For such striking effects of these perturbations on DNMT1 chromatin association, it is surprising that the effects on DNA methylation are subtle (Figure 7). This brings into an important question that to what extent does PAF15Ub2 support DNA methylation maintenance? Does it really matter in a physiologic context and for the faithful propagation of DNA methylation patterns through mitotic cell divisions? To strengthen the impact of this work, it will be important to expand DNA methylation analyses and extend findings from *Xenopus* extracts to mammalian cells.

This point is important and well taken. According to the reviewer’s suggestion, we have now included experiments using mouse ES cells (mESCs) showing that the mutagenesis of the USP7 binding sequences in PAF15 using genome editing resulted in (1) decreased USP7 binding ability of PAF15, (2) increased PAF15 chromatin binding levels, and (3) increased DNA methylation levels at LINE-1 element in mESCs. These results provide further evidence to support our conclusion that USP7 regulates PAF15 chromatin unloading and prevents unscheduled hypermethylation. We describe these results in the revised Figure 8A-C and discussed in the main text (pages 20-21).

[Editors’ note: further revisions were suggested prior to acceptance, as described below.]

1. Data in Figure 7A shows disruption of USP7 alone is not sufficient to disrupt the rate of DNA methylation maintenance.

We have already described that immunodepletion of USP7 does not increase the rate of DNA methylation maintenance in *Xenopus* egg extracts, probably due to the unloading of PAF15/PCNA complex by ATAD5 in late S phase. However, we have now shown that the inhibition of PAF15-USP7 interaction caused the significant increase in LINE-1 DNA methylation in both mESCs and mEpiLCs. Please see the response to reviewer’s comment #3.

Data in Figure 7 cannot claim that the observed increase in DNA methylation in USP7/ATAD5-depleted extracts is related to PAF15.

This point is important and well taken. As we previously reported, PAF15 depletion induces an alternative pathway via UHRF1-dependent histone H3 ubiquitylation. This makes it difficult to directly test the requirement of PAF15 for increased DNA methylation in USP7/ATAD5-depleted extracts. According to reviewer’s comment, we performed DNMT1 immunoprecipitation from chromatin lysates to investigate the DNMT1-PAF15Ub2 or H3Ub2 interaction in the absence of USP7 and ATAD5. Our result shows that USP7/ATAD5 depletion actually increased the level of PAF15Ub2, but not H3Ub2, in the DNMT1 immunoprecipitates. This data supports our idea that the PAF15Ub2-DNMT1 complex is responsible for increased DNA methylation in USP7/ATAD5-depleted extracts (shown in the revised Figure 7—figure supplement1D).

Furthermore, the rate of DNA replication in these doubly deleted extracts also looks faster than mock depletion (Supp Figure 7B).

This point is important and well taken. According to the reviewer’s comment, we have repeated the experiment and compared DNA replication in mock-, USP7-, ATAD5- or ATAD5/USP7-depleted extracts under the same experimental conditions. This experiment confirmed that USP7 and ATAD5 double depletion showed little effect on DNA replication, although ATAD5-depletion alone resulted in slightly increased DNA replication (shown in revised Figure 7—figure supplement1B).

2. The new DNA methylation analysis in Figure 8C analyzing mESCs with disrupted PAF15-USP7 interaction does not show statistically significant differences in EpiLCs.

This point is important and well taken. According to the reviewer’s suggestion, we have repeated experiments using mouse epiblast like cells, which were harvested at 72 hr after cells were in the epiblast medium to complete the differentiation. Our new result clearly demonstrates that expression of PAF15 mutant in mEpiLCs resulted in increased DNA methylation levels at LINE-1 element as well as in mESCs. (shown in revised Figure 8C).

3. The analysis of newly performed ITC data lacks sufficient clarity. The ITC curve for PAF15 WT in Figure 2—figure supplement 1 does not appear to reflect a 1:1 stoichiometric binding. What was the N value when the Kd of 32.7 µM was determined? If the N value is substantially deviated from 1, an explanation would be warranted. In addition, in the method section, should "26/600 HiLoad 26/600 Superdex 75" be "HiLoad 26/600 Superdex 75"?

The interaction between USP7 UBL domains and PAF15 was relatively weak, and the titration curve did not show a sigmoid. Due to the limitation of the solubility of the proteins, we performed the ITC assay with the protein concentrations described in the previous version of manuscript. However, the excess molar ratio titration of ligands to the protein is recommended by the manufacturer when binding affinity is low, and such a titration method can sometimes be employed for weak interaction (Abe et al., Science 2018; Zaidi et al., PLOS ONE 2013; Ciulli Methods Mol Biol 2015). In this case, the stoichiometry *N* is apparent value, but the binding stoichiometry between USP7 and PAF15 was 1.23 ± 0.21. We added the apparent N value in Figure 2—figure supplement 1. We also thank you for pointing the mislabeling of the SEC column. We have amended it according to the reviewer’s suggestion.